# MIND THE PAD – CNNS CAN DEVELOP BLIND SPOTS

**Bilal Alsallakh**
Facebook AI

**Narine Kokhlikyan**
Facebook AI

**Vivek Miglani**
Facebook AI

**Jun Yuan**
NYU

**Orion Reblitz-Richardson**
Facebook AI

## ABSTRACT

We show how feature maps in convolutional networks are susceptible to spatial bias. Due to a combination of architectural choices, the activation at certain locations is systematically elevated or weakened. The major source of this bias is the padding mechanism. Depending on several aspects of convolution arithmetic, this mechanism can apply the padding unevenly, leading to asymmetries in the learned weights. We demonstrate how such bias can be detrimental to certain tasks such as small object detection: the activation is suppressed if the stimulus lies in the impacted area, leading to blind spots and misdetection. We propose solutions to mitigate spatial bias and demonstrate how they can improve model accuracy.

## 1 MOTIVATION

Convolutional neural networks (CNNs) serve as feature extractors for a wide variety of machine-learning tasks. Little attention has been paid to the spatial distribution of activation in the feature maps a CNN computes. Our interest in analyzing this distribution is triggered by mysterious failure cases of a traffic light detector: The detector successfully detects a small but visible traffic light in a road scene. However, it fails completely in detecting the same traffic light in the next frame captured by the ego-vehicle. The major difference between both frames is a limited shift along the vertical dimension as the vehicle moves forward. Therefore, the drastic difference in object detection is surprising given that CNNs are often assumed to have a high degree of translation invariance [8; 17].

The spatial distribution of activation in feature maps varies with the input. Nevertheless, by closely examining this distribution for a large number of samples, we found consistent patterns among them, often in the form of artifacts that do not resemble any input features. This work aims to analyze the root cause of such artifacts and their impact on CNNs. We show that these artifacts are responsible for the mysterious failure cases mentioned earlier, as they can induce 'blind spots' for the object detection head. Our contributions are:

- Demonstrating how the padding mechanism can induce spatial bias in CNNs (Section 2).
- Demonstrating how spatial bias can impair downstream tasks (Section 3).
- Identifying uneven application of 0-padding as a resolvable source of bias (Section 5).
- Relating the padding mechanism with the foveation behavior of CNNs (Section 6).
- Providing recommendations to mitigate spatial bias and demonstrating how this can prevent blind spots and boost model accuracy.

## 2 THE EMERGENCE OF SPATIAL BIAS IN CNNS

Our aim is to determine to which extent activation magnitude in CNN feature maps is influenced by location. We demonstrate our analysis on a publicly-available traffic-light detection model [36]. This model implements the SSD architecture [26] in TensorFlow [1], using MobileNet-v1 [13] as a feature extractor. The model is trained on the BSTLD dataset [4] which annotates traffic lights in road scenes. Figure 1 shows two example scenes from the dataset. For each scene, we show two feature maps computed by two filters in the 11[th] convolutional layer. This layer contains 512 filters whose feature maps are used directly by the first box predictor in the SSD to detect small objects.

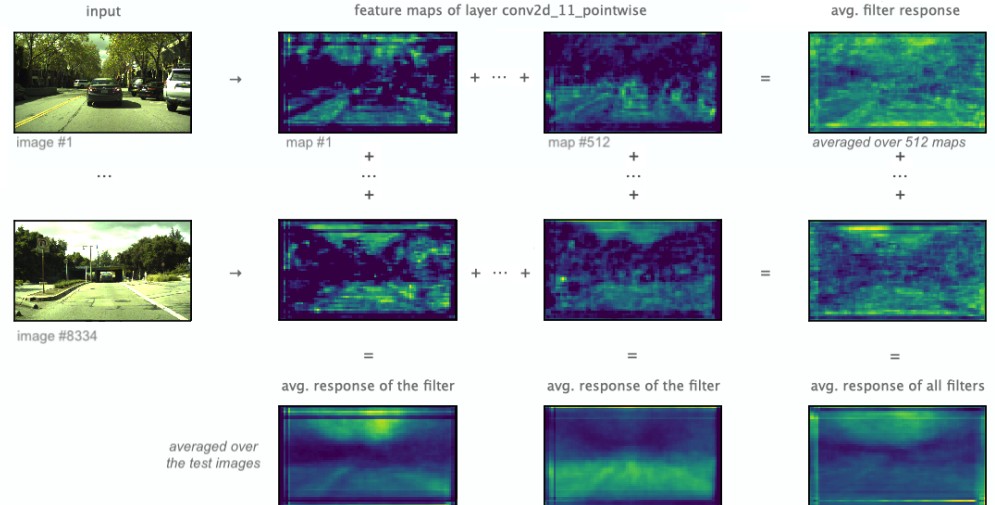

Figure 1: Averaging feature maps per input (column marginal) and per filter (row marginal) in the last convolutional layer of a traffic light detector. Color indicates activation strength (the brighter, the higher), revealing line artifacts in the maps. These artifacts are the manifestation of spatial bias.

The bottom row in Figure 1 shows the average response of each of the two aforementioned filters, computed over the test set in BSTLD. The first filter seems to respond mainly to features in the top half of the input, while the second filter responds mainly to street areas. There are visible lines in the two average maps that do not seem to resemble any scene features and are consistently present in the individual feature maps. We analyzed the prevalence of these line artifacts in the feature maps of all 512 filters. The right column in Figure 1 shows the average of these maps per scene, as well as over the entire test set (see supplemental for all 512 maps). The artifacts are largely visible in the average maps, with variations per scene depending on which individual maps are dominant.

A useful way to make the artifacts stand out is to neutralize scene features by computing the feature maps for a zero-valued input. Figure 2 depicts the resulting average map for each convolutional layer after applying ReLU units. The first average map is constant as we expect with a 0-valued input. The second map is also constant except for a 1-pixel boundary where the value is lower at the left border and higher at the other three borders. We magnify the corners to make these deviations visible. The border deviations increase in thickness and in variance at subsequent layers, creating multiple line artifacts at each border. These artifacts become quite pronounced at `ReLU 8` where they start to propagate inwards, resembling the ones in Figure 1.

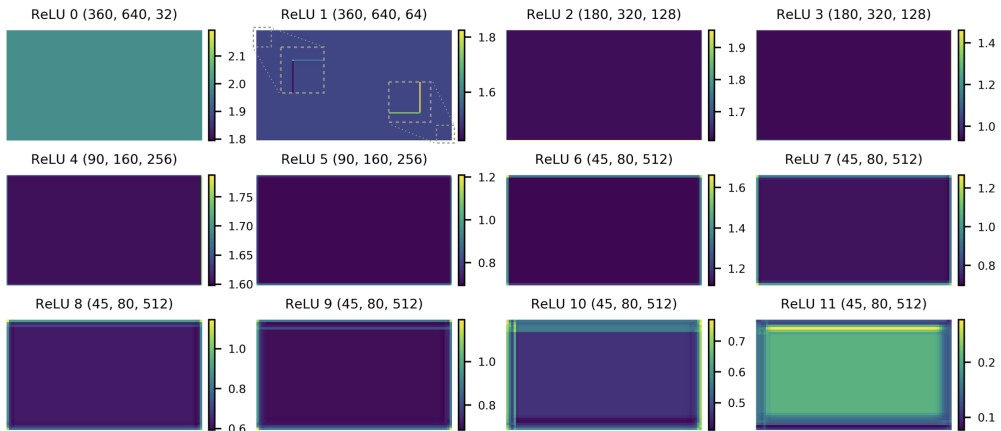

Figure 2: Activation maps for a 0 input, averaged over each layer's filters (title format: H×W×C).

It is evident that the 1-pixel border variations in the second map are caused by the padding mechanism in use. This mechanism pads the output of the previous layer with a 1-pixel 0-valued border in order to maintain the size of the feature map after applying 3x3 convolutional. The maps in the first layer are not impacted because the input we feed is zero valued. Subsequent layers, however, are increasingly impacted by the padding, as preceding bias terms do not warrant 0-valued input.

It is noticeable in Figure 2 that the artifacts caused by the padding differ across the four borders. To investigate this asymmetry, we analyze the convolutional kernels (often called filters) that produce the feature maps. Figure 3 depicts a per-layer mean of these 3x3 kernels. These mean kernels exhibit different degrees of asymmetry in the spatial distribution of their weights. For example, the kernels in `L1` assign (on average) a negative weight at the left border, and a positive weight at the bottom. This directly impacts the padding-induced variation at each border. Such asymmetries are related to uneven application of padding as we explain in Section 5.

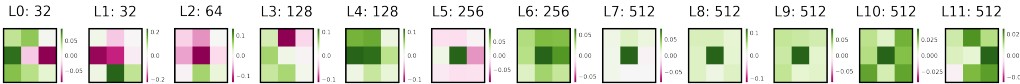

Figure 3: Mean kernel per convolutional layer. All kernels are $3 \times 3$, the titles show their counts.

## 3 IMPLICATIONS OF SPATIAL BIAS

We demonstrate how feature-map artifacts can cause blind spots for the SSD model. Similar issues arise in several small-object detectors, e.g., for faces and masks, as well as in pixel-oriented tasks such as semantic segmentation and image inpainting (see supplemental for examples).

Figure 4 illustrates how the SSD predicts small objects based on the feature maps of the 11-th convolutional layer. The SSD uses the pixel positions in these maps as anchors of object proposals. Each proposal is scored by the SSD to represent a target category, with "background" being an implicit category that is crucial to exclude irrelevant parts of the input. In addition to these scores, the SSD computes a bounding box to localize the predicted object at each anchor. We examine

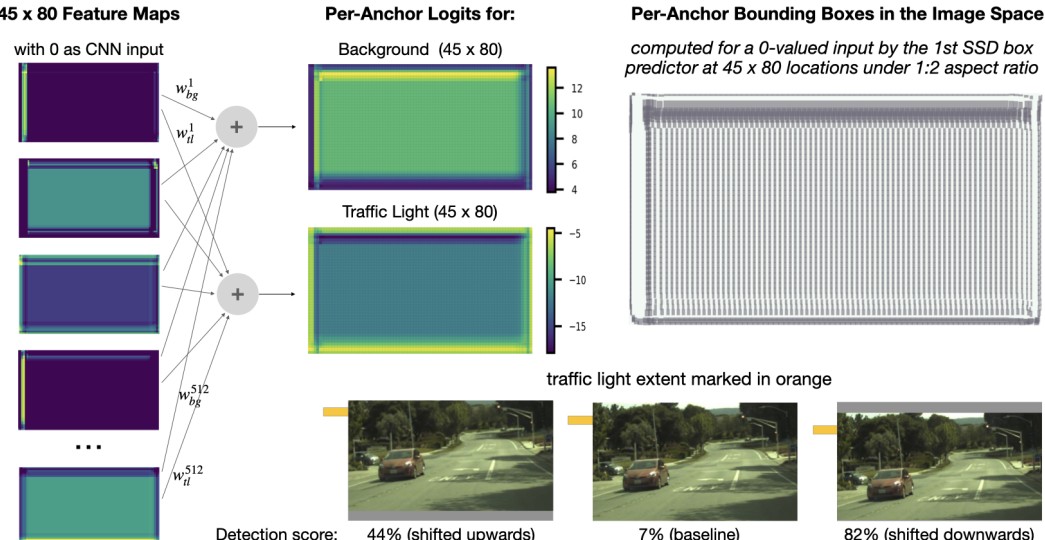

Figure 4: The formation of blind spots in SSD, illustrated via its box predictor internals with a zero-valued input. The predictor uses spatial anchors to detect and localize the target object at $45 \times 80$ possible locations based on 512 feature maps. Certain anchors are predisposed to predict `background` due to feature-map artifacts, as evident in the logit maps. Traffic lights at the corresponding location cannot be detected as demonstrated with a real scene (middle one in the bottom).

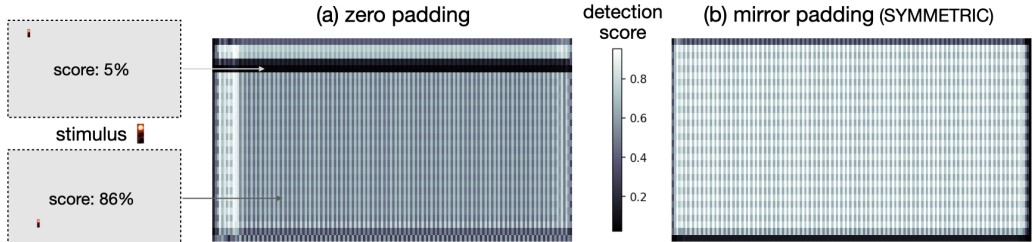

Figure 5: (a) A map showing via color the detection score the SSD computes for a traffic light when present at various locations. The detection is muted when the stimulus lies in the area impacted by the artifacts. (b) The same map after changing the padding method to SYMMETRIC. The detection scores are rather constant except for periodic variations due to the SSD's reliance on anchors.

object proposals computed at 1:2 aspect ratio, as they resemble the shape of most traffic lights in the dataset. We visualize the resulting score maps both for the background category and for traffic lights, when feeding a 0-valued input to the SSD. We also visualize the bounding boxes of these proposals in the image space. The SSD predicts the image content to be of background category at all anchor locations, as evident from the value range in both score maps. Such predictions are expected with an input that contains no traffic lights. However, the line artifacts in the feature maps have a strong impact on the score maps. These artifacts elevate the likelihood of anchors closer to the top to be classified as background (see the yellow band in the background score map). Conversely, these anchors have significantly lower scores for the traffic light category, compared with other anchors in the feature map. Such difference in the impact on the target categories is due to the different weights the SSD assigns to the feature maps for each target. As a result, the artifacts lead to potential blind spots in which the scores for certain categories are artificially muted.

To validate whether or not the blind spots hinder object detection, we examine road scenes that contain highly-visible traffic light instances in the impacted area. Figure 4-bottom shows an example of such a scene. The SSD computes a low detection score of 7% when the traffic light lies in the blind spot (see middle image), far below the detection false-positive cutoff. Shifting the scene image upwards or downwards makes the instance detectable with a high score as long as it lies outside the blind spot. This explains the failure cases mentioned in Section 1. To further validate this effect, we run the SSD on baseline images that each contains one traffic light instance at a specific location in the input. We store the detection score for each instance. Figure 5a depicts the computed scores in a 2D map. It is evident that the model fails to detect the traffic light instance exactly when it is located within the "blind spot" band. The artifacts further disrupt the *localization* of the objects as evident in the top-right plot in Figure 4 which shows per-anchor object proposals computed for a 0 input.

## 4 REMINDER: WHY IS PADDING NEEDED IN CNNS?

Padding is applied **at most convolutional layers** in CNNs to serve two fundamental purposes:

**Maintaining feature map size** A padding that satisfies this property is often described as SAME or HALF padding. FULL padding expands the maps by kernel size - 1 along each dimension. VALID padding performs no padding, eroding the maps by the same amount. SAME padding is important to (1) design deep networks that can handle arbitrary input size (a challenge in the presence of gradual erosion), (2) maintain the aspect ratio of non-square input, and (3) concatenate feature maps from different layers as in Inception [39] and ResNet [12] models.

**Reducing information bias against the boundary** Consider a $3 \times 3$ kernel applied to a 2D input. An input location at least 2 pixels away from the boundary contributes to nine local convolution operations when computing the feature map. On the other hand, the corner is involved only one time under VALID padding, four times under a 1-pixel SAME 0-padding, and nine times under a 2-pixel FULL 0-padding. With SAME 0-padding, the cumulative contribution differences among the input pixels grow exponentially over the CNN layers. We refer to such uneven treatment of input pixels as the *foveation behavior* of the padding mechanism and elaborate on this in Section 6.

We next explore solutions to the issues that cause padding to induce spatial bias.

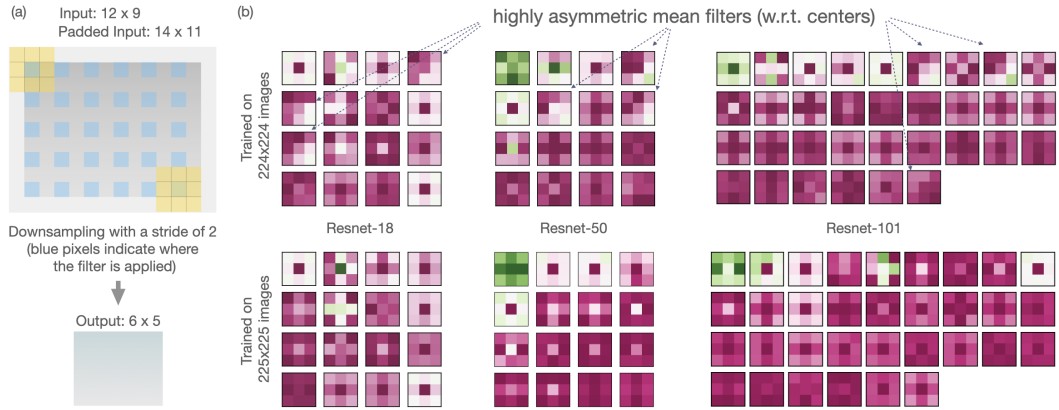

Figure 6: (a) Illustrating the problem of uneven padding when down-sampling at a stride of 2. The padding along x-axis is consumed only at the left side. (b) Mean $3 \times 3$ filters in three ResNet models, trained on ImageNet with two input sizes. Color encodes average weight (green is positive). A size that induces uneven padding (top row) can lead to asymmetries, esp. around down-sampling layers. These asymmetries are mitigated when the input size induces no uneven padding (bottom row).

## 5 ELIMINATING UNEVEN APPLICATION OF PADDING

While useful to reduce bias against the boundary, applying padding at down-sampling layers can lead to asymmetry in CNN internals. Figure 6a illustrates the source of this asymmetry when strided convolution is used for downsampling: At one side of the feature map, the padding is consumed by the kernel while at the other side it is not. To warrant even application of padding throughout the CNN, the following must hold at all $d$ down-sampling layers, where $(h_i, w_i)$ is the output shape at the i-th layer with $k_i^h \times k_i^w$ as kernel size, $(s_i^h, s_i^w)$ as strides, and $= (p_i^h, p_i^w)$ as padding amount (refer to appendix A for a proof):

$$\forall i \in \{1, \ldots, d\} : h_{i-1} = s_i^h \cdot (h_i - 1) + k_i^h - 2 \cdot p_i^h \quad \wedge \quad w_{i-1} = s_i^w \cdot (w_i - 1) + k_i^w - 2 \cdot p_i^w \quad (1)$$

The values $h_0$ and $w_0$ represent the CNN input dimensions. The above constraints are not always satisfied during training or inference with arbitrary input dimensions. For example, ImageNet classifiers based on ResNet [12] and MobileNet [13] contain five down-sampling layers ($d = 5$) that apply 1-pixel 0-padding before performing 2-strided convolution. To avoid uneven application of padding, the input to these CNNs must satisfy the following, as explained in appendix A:

$$h_0 = a_1 \times 2^d + 1 = 32 \cdot a_1 + 1 \quad \text{and} \quad w_0 = a_2 \times 2^d + 1 = 32 \cdot a_2 + 1 \quad \text{where} \quad a_1, a_2 \in \mathbb{N}^+ \quad (2)$$

The traditional [1] and prevalent input size for training ImageNet models is $224 \times 224$. This size violates Eq. 2, leading to uneven padding at every down-sampling layer in ResNet and MobileNet models where 0-padding is effectively applied only at the left and top sides of layer input. This over-represents zeros at the top and left sides of $3 \times 3$ feature-map patches the filters are convolved with during training. The top row of Figure 6b shows per-layer mean filters in three ResNet models in PyTorch [33], pre-trained on ImageNet with $224 \times 224$ images. In all of these models, a few of the mean filters, adjacent to down-sampling layers, exhibit stark asymmetry about their centers.

We increase the image size to $225 \times 225$ without introducing additional image information[2]. This size satisfies Eq. 2, warranting even application of padding at every downsampling layer in the above models. Retraining the models with this size strongly reduces this asymmetry as evident in the bottom row of Figure 6b. This, in turn, visibly boosts the accuracy in all models we experimented with as we report in Table 1. The accuracy did not improve further when we retrained two of the models, ResNet-18 and ResNet-34, on $226 \times 226$ images. This provides evidence that the boost is due to eliminating uneven padding and not merely due to increasing the input size.

---

[1] This size has been used to facilitate model comparison on ImageNet, since the inception of AlexNet.

[2] This is done via constant padding. The side to pad with one pixel is chosen at random to balance out the application of padding at both sides over the training set. No additional padding is applied at further layers.

Replacing 0-padding with a padding method that reuses feature map values can alleviate the asymmetry in the learned filters in the presence of unevenly applied padding. Another possibility is to use a rigid downsampling kernel, such as max-pooling, instead of a learned one. Appendix C demonstrates both possibilities. Finally, antialiasing before downsampling [43] can strongly reduce the asymmetry as we elaborate in Section 8 and in Appendix E.

Table 1: Top-1 (and top-5) accuracy of five ImageNet classifiers trained with different input sizes.

| Input Size $^2$ | MobileNet | ResNet-18 | ResNet-34 | ResNet-50 | ResNet-101 |
|---|---|---|---|---|---|
| 224×224 | 68.19 (88.44) | 69.93 (89.22) | 73.30 (91.42) | 75.65 (92.47) | 77.37 (93.56) |
| 225×225 | 68.80 (88.78) | 70.27 (89.52) | 73.72 (91.58) | 76.01 (92.90) | 77.67 (93.81) |

Even when no padding is applied ($p_i^h = 0$ or $p_i^w = 0$), an input size that does no satisfy Eq. 1 can lead to uneven *erosion* of feature maps, in turn, reducing the contribution of pixels from the impacted sides (Fig 7e. Satisfying Eq 1 imposes a restriction on input size, e.g., to values in increments of $2^d = 32$ with the above models (193×193, 225×225, 257×257, ...). Depending on the application domain, this can be guaranteed either by resizing an input to the closest increment, or by padding it accordingly with suited values.

## 6    Padding Mechanism and Foveation

By foveation we mean the unequal involvement of input pixels in convolutional operations throughout the CNN. Padding plays a fundamental role in the foveation behavior of CNNs. We visualize this behavior by means of a *foveation map* that counts for each input pixel the number of convolutional paths through which it can propagate information to the CNN output. We obtain these counts by computing the *effective receptive field* [28] for the sum of the final convolutional layer after assigning all weights in the network to 1 (code in supplemental). Neutralizing the weights is essential to obtain per-pixel *counts* of input-output paths that reflect the foveation behavior.

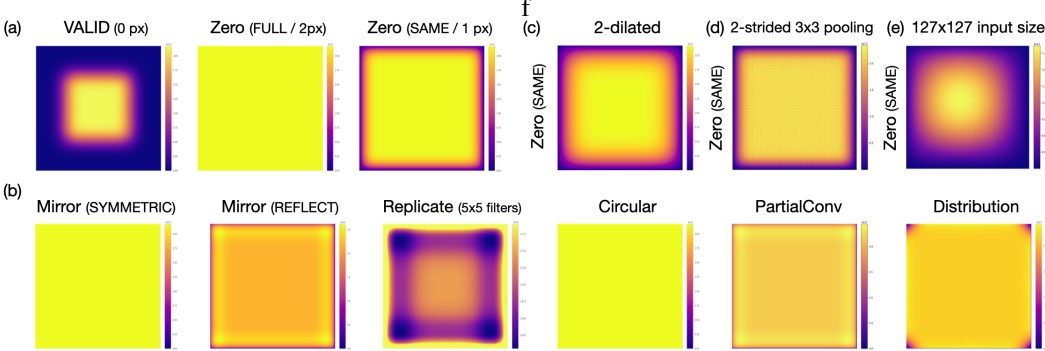

Figure 7: Foveation behavior of different padding methods applied to VGG-19 [37], and illustrated in a 512 × 512 input space (unless otherwise stated). Color represents the number of paths to the output for each input pixel. (a) The difference between VALID, FULL, and SAME 0-padding. (b) SAME alternatives to 0-padding. (c) Dilation amplifies foveation of SAME 0-padding. (d) Strides can lead to checkerboard patterns. (e) Foveation effects are more extensive in smaller inputs (relative to input size) and are sensitive to uneven padding.

Figure 7a shows the extensive foveation effect when no padding is applied. The diminishing contribution of vast areas of the input explains the drastic drop in accuracy recently observed under VALID padding [16]. In contrast, FULL 0-padding does not incur foveation, however, at the cost of increasing the output size after each layer, making it impractical as explained in Section 4. SAME 0-padding incurs moderate foveation at the periphery, whose absolute extent depends on the number of convolutional layers and their filter sizes. Its *relative* extent depends on the input size: the larger the input, the larger the ratio of the constant area in yellow (refer to appendix B for a detailed example).

Figure 7b shows the foveation behavior of alternatives to SAME 0-padding that have roots in wavelet analysis [19] and image processing [27]. **Mirror padding** mirrors pixels at the boundary to fill the padding area. When the border is included (SYMMETRIC mode in TensorFlow) all input pixels have an equal number of input-output paths [3], resulting in a uniform foveation map. When the border is not included (REFLECT mode both in PyTorch and in TensorFlow), the map exhibits bias against the border and towards a contour in its proximity. This bias is amplified over multiple layers. **Replication padding** exhibits the opposite bias when the padding area is wider than 1 pixel. This is because it replicates the outer 1-pixel border multiple times to fill this area [3]. The method is equivalent to SYMMETRIC if the padding area is 1-pixel wide. **Circular padding** wraps opposing borders, enabling the kernels to seamlessly operate on the boundary and resulting in a uniform map. **Partial Convolution** [22] has been proposed as a padding method that treats pixels outside the original image as missing values and rescales the computed convolutions accordingly [23]. Its foveation behavior resembles reflective padding [3]. **Distribution padding** [30] resizes the input to fill the padding area around the original feature map, aiming at preserving the distribution of the map. Its foveation map is largely uniform, except for the corners and edges.

**Impact of input size**  Besides influencing the relative extent of foveation effects, the input size also determines the presence of uneven padding (or uneven feature-map erosion), as we discussed in Section 5. Figure 7e shows the foveation map for VGG-19 with a $127 \times 127$ input. This input violates Eq. 1 at every downsampling layer (appendix A), leading to successive feature map erosion at the bottom and right sides which is reflected in the foveation map (see appendix B for a detailed example). The bottom-right part of the input is hence less involved in the CNN computations.

**Impact of dilation**  We assign a dilation factor of 2 to all VGG-19 convolutional layers. While this exponentially increases the receptive field of the neurons at deeper layers [42], dilation doubles the extent of the non-uniform peripheral areas that emerge with SAME 0-padding as evident in Figure 7c. SYMMETRIC and circular padding maintain uniform foveation maps regardless of dilation [3]. In contrast, dilation increases the complexity of these maps for REFLECT and replication padding.

**Impact of strides**  Whether learned on based on pooling, downsampling layers can amplify the impact of succeeding convolutional layers on foveation behaviour. Furthermore, these layers can cause input pixels to vary in the count of their input-output paths. This can happen when the kernel size is not divisible by the stride, leading to a checkerboard pattern in the foveation maps. This manifests in ResNet models as we illustrate in appendix B. In VGG-19, all max-pooling layers use a stride of 2 and kernel size of 2. Changing the kernel size to 3 leads to a checkerboard pattern as evident in Figure 7d. Such effects were shown to impact pixel-oriented tasks [32].

The padding technique and its foveation behaviour have direct impact on feature-map artifacts (Section 7), and on the ability of CNNs to encode spatial information (Section 8). Understanding the foveation behavior is key to determine how suited a padding method is for a given task. For example, small object detection is known to be challenging close to the boundary [26], in part due to the foveation behavior of SAME 0-padding. In Figure 5b, we change the padding method in the SSD to SYMMETRIC. The stimulus is noticeably more detectable at the boundary, compared with 0-padding [4]. In contrast, ImageNet classification is less sensitive to foveation effects because the target objects are mostly located away from the periphery. Nevertheless, the padding method was shown to impact classification accuracy [23] because it still affects feature map artifacts.

## 7 PADDING METHODS AND FEATURE MAP ARTIFACTS

It is also noticeable that the score map in Figure 5b is more uniform than in Figure 5a. In particular, under SYMMETRIC padding the model is able to detect traffic lights placed in the blind spots of the original 0-padded model. To verify whether the line artifacts in Figure 2 are mitigated, we inspect the mean feature maps of the adapted model. With a constant input, SYMMETRIC padding warrants constant maps throughout the CNN because it reuses the border to fill the padding area. Instead, we average these maps over 30 samples generated uniformly at random. Figure 8 depicts the mean maps which are largely uniform, unlike the case with 0-padding.

---

[3] Refer to appendix F or to `http://mind-the-pad.github.io` for visual illustration and further theoretical analysis of the foveation behavior.

[4] Since the input size causes uneven application of padding, the right and bottom borders are still challenging.

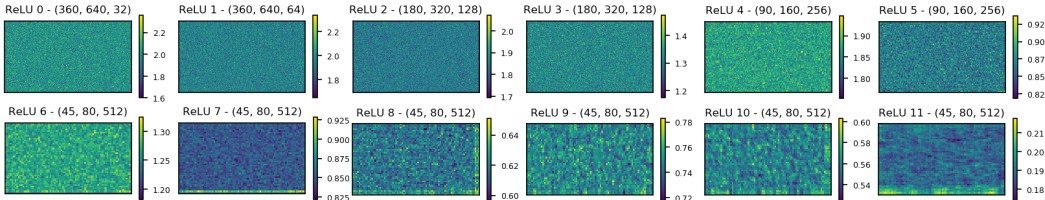

Figure 8: The same feature maps in Figure 2, generated under mirror padding and averaged over 30 randomly-generated input samples. The line artifacts induced by 0-padding are largely mitigated.

To further analyze the impact of SYMMETRIC padding, we retrain the adapted model following the original training protocol. This significantly improves the average precision (AP) as reported in Table 2 under different overlap thresholds (matching IoU), confirming that small object detection is particularly sensitive to feature-map artifacts.

Table 2: Performance of the SSD traffic light detector, trained under two different padding schemes.

| Average Precision (AP) | AP@.20IOU | AP@.50IOU | AP@.75IOU | AP@.90IOU |
|---|---|---|---|---|
| Zero Padding | 80.24% | 49.58% | 3.7% | 0.007% |
| Mirror Padding | 83.20% | 57% | 8.44% | 0.02% |

Of the padding methods listed in Section 6, mirror padding in both SYMMETRIC and REFLECT modes, PartialConv, and circular padding are generally effective at reducing feature map artifacts that emerge under zero padding, in particular salient line patterns. In contrast, distribution padding can induce significant artifacts. Refer to appendix D for comparative examples of artifacts under the aforementioned padding schemes.

**Artifact magnitude and propagation**  While feature-map artifacts are induced by the padding mechanism at the boundary, their magnitude and inward propagation in the maps are impacted by several architectural aspects of CNNs. In particular, certain normalization schemes such as batchnorm [15] tend to limit the range of variation within a feature map and to relatively harmonize this range across different maps. This, in turn, impacts how possible artifacts in these maps accumulate when they are processed by the next convolutional layer. Similarly, artifacts that manifest after applying ReLU units are of a positive sign. These factors were instrumental in the formation of potential blind spots described in Section 3. We hence recommend to involve non-convolutional layers when inspecting the feature maps. Besides having possible impact on artifact magnitude, several aspects of convolution arithmetic, such as filter size and dilation factors, can also impact the spatial propagation of these artifacts.

## 8 RELATED FINDINGS AND TAKEAWAYS

Handling the boundary is an inherent challenge when dealing with spatial data [9]. Mean padding is known to cause visual artifacts in traditional image processing, with alternative methods proposed to mitigate them [24]. CNNs have been often assumed to deal with such effects implicitly. Innamorati et al [14] propose learning separate sets of filters dedicated to the boundaries to avoid impacting the weights learned by regular filters. A grouped padding strategy, proposed to support $2 \times 2$ filters [41], offers avenues to mitigate uneven padding and corresponding skewness in foveation maps without restrictions on input size (see our note in appendix B for explanation). Finally, insights from signal and image processing [10; 11] could inspire further CNN padding schemes.

Zero padding has been recently linked to CNNs' ability to encode position information [7; 16; 18; 29]. In contrast, circular padding was shown to limit this ability [7] and to boost shift invariance [35]. The input sizes in those studies do induce uneven padding. This can be, in part, the underlying mechanism behind the aforementioned ability. Whether or not this ability is desirable depends on the task, with several methods proposed to explicitly encode spatial information [5; 6; 20; 25; 29; 31].

Downsampling using max-pooling or strided convolution has been shown to impact shift invariance in CNNs by incurring aliasing effects [3; 38; 43]. These effects can manifest in the same symptoms we reported in Section 1, albeit for a different reason. Zhang [43] demonstrated how blurring the feature maps before subsampling mitigates aliasing effects and improves ImageNet classification accuracy of various popular CNNs. We analyzed the mean filters in antialiased MobileNet and ResNet models pre-trained on ImageNet under 0-padding, with $224\times224$ as input size (refer to Appendix E). We found that antialiasing can also mitigate the asymmetry of mean filters that exhibited high asymmetry in the baseline models, especially at deeper layers. This is remarkable given that these models are trained on $224\times224$ images, which incurs one-sided zero padding at every downsampling layer. This could, in part, be attributed to the ability of the BlurPool operator used in antialiased CNN to smoothen the acuity of zero-padded borders, in turn, reducing the value imbalance incurred by one-sided padding. Further analysis is needed to examine the interaction between padding and aliasing effects in CNNs and to establish possible synergy between antialiasing and eliminating uneven application of padding.

Luo et al [28] drew connections between effective receptive fields and foveated vision. Our analysis links foveation behavior with the padding scheme and suggests that it might occur implicitly in CNNs when using VALID or SAME 0-padding, without the need for explicit mechanisms [2; 21]. Furthermore, it explains the drastic accuracy drop noted by [16] under VALID padding, which is amplified by feature map erosion.

**Choosing a padding method**    SAME 0-padding is by far the most widely-used method. Compared with other methods, it can enable as much as $50\%$ faster training and inference. Problem-specific constraints can dictate different choices [34; 35; 40]. In the lack of a universally superior padding method, we recommend considering multiple ones while paying attention to the nature of the data and the task, as well as to the following aspects:

- Feature-map statistics: 0-padding can alter the value distribution within the feature maps and can shift their mean value in the presence of ReLU units. The alternatives presented in Section 6 tend to preserve this distribution, thanks to reusing existing values in the maps.
- Foveation behavior: 0-padding might not be suited for tasks that require high precision at the periphery, unlike circular and SYMMETRIC mirror padding.
- Interference with image semantics (esp. with a padding amount $> 1$ pixel): For example, circular padding could introduce border discontinuities unless the input is panoramic [35].
- Potential to induce feature map artifacts: All alternatives to 0-padding induce relatively fewer artifacts, except for Distribution padding [30] (see appendix D).

We also recommend eliminating uneven padding at downsampling layers both at training and at inference time, as we illustrated in Section 5. This is especially important when zero padding is applied and the downsampling is learned. The scripts used to generate the visualizations in this paper are available in the supplemental as well as at http://mind-the-pad.github.io.

**Summary**    We demonstrated how the padding mechanism can induce spatial bias in CNNs, in the form of skewed kernels and feature-map artifacts. These artifacts can be highly pronounced with the widely-used 0-padding when applied unevenly at the four sides of the feature maps. We demonstrated how such uneven padding can inherently take place in state-of-the-art CNNs, and how the artifacts it causes can be detrimental to certain tasks such as small object detection. We provided visualization methods to expose these artifacts and to analyze the implication of various padding schemes on boundary pixels. We further proposed solutions to eliminate uneven padding and to mitigate spatial bias in CNNs. Further work is needed to closely examine the implications of spatial bias and foveation in various applications (see supplementary for examples), as well as padding impact on recurrent models and 1-D CNNs.

## ACKNOWLEDGEMENT

We are thankful to Ross Girshick for providing useful recommendations and experiment ideas, and to Shubham Muttepawar for implementing an interactive tool out of our analysis scripts, guided by our front-end specialist Edward Wang and our AI user-experience designer Sara Zhang.

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

## A    ELIMINATING UNEVEN APPLICATION OF PADDING

Consider a CNN with $d$ downsampling layers, $L_1, L_2, ..., L_d$. To simplify the analysis and without loss of generality we assume that the kernels in these layers are of square shape and that all other layers maintain their input size. We denote by $s_i$ and $k_i$ the stride and kernel size of layer $L_i$. We denote by $h_i$ and $w_i$ the dimensions of the feature maps computed by $L_i$. We denote by $h_0$ and $w_0$ the size of the CNN input. We examine the conditions to warrant no uneven application of padding along the height dimension. Parallel conditions apply to the width dimension.

We denote by $\bar{h}_i$ the height of the padded input to $L_i$. The effective portion $\hat{h}_i \leq \bar{h}_i$ of this amount processed by the convolutional filters in $L_i$ is equal to:

$$\hat{h}_i = s_i \cdot (h_i - 1) + k_i$$

Our goal is to warrant that $\hat{h}_i = \bar{h}_i$ to prevent information loss and to avoid uneven padding along the vertical dimension when the unconsumed part $\bar{h}_i - \hat{h}_i < s_i$ is an odd number.

Since the non-downsampling layers maintain their input size, we can formulate the height of the padded input as follows:

$$\bar{h}_i = h_{i-1} + 2 \cdot p_i$$

where $p_i$ is the amount of padding applied at the top and at the bottom of the input in $L_i$. Accordingly, we can warrant no uneven padding if the following holds:

$$\forall i \in [1..d]: \quad h_{i-1} = s_i \cdot (h_i - 1) + k_i - 2 \cdot p_i \tag{3}$$

**Example 1: ResNet-18**    This network contains five downsampling layers ($d = 5$) all of which use a stride of 2. Despite performing downsampling, all of these layers apply a padding amount entailed by SAME padding to avoid information bias against the boundary. In four of these layers having $3 \times 3$ kernels ($k_i = 3$), the amount used is $p_i = 1$. For the first layer having $7 \times 7$ kernels, this amount is equal to 3. In both cases, the term $k_i - 2 \cdot p_i$ in Eq. 3 is equal to 1. To warrant no uneven padding along the vertical dimension, the heights of the feature maps at downsampling layers should hence satisfy:

$$\forall i \in [1..d]: \quad h_{i-1} = 2 \cdot (h_i - 1) + 1 = 2 \cdot h_i - 1$$

Accordingly, the input height should satisfy:

$$h_0 = 2^d \cdot h_d - (2^d - 1) = 2^d \cdot (h_d - 1) + 1$$

where $h_d$ is the height of the final feature map, and can be any natural number larger than 1 to avoid a degenerate case of a $1 \times 1$ input. The same holds for the input width:

$$w_0 = 2^d \cdot (w_d - 1) + 1$$

A $225 \times 225$ input satisfies these constraints since $225 = 2^5 \cdot 7 + 1$, yielding even padding in all five downsampling layers and output feature maps of size $8 \times 8$.

**Example 2: VGG-16**    This network contains five max-pooling layers ($d = 5$) all of which use a stride of 2 and a kernel size of 2 and apply no padding. To warrant no uneven padding along the vertical dimension, the heights of the feature maps at all of these layers should hence satisfy:

$$\forall i \in [1..d]: \quad h_{i-1} = 2 \cdot (h_i - 1) + 2 = 2 \cdot h_i$$

Accordingly, the input dimensions should satisfy:

$$h_0 = 2^d \cdot h_d \quad \text{and} \quad w_0 = 2^d \cdot w_d \tag{4}$$

A $224 \times 224$ input satisfies these constraints since $224 = 2^5 \cdot 7$, causing no feature-map erosion at any downsampling layer and resulting in output feature maps of size $7 \times 7$.

## B    THE EXTENT OF FOVEATION UNDER SAME 0-PADDING

We illustrate how the absolute extent of foveation under SAME 0-padding depends on the number of convolutional layers, and how its relative extent depends on the input size.

In the following maps, color represents the number of paths to the CNN output for each input pixel. Note: The checkerboard pattern is caused by downsampling layers in ResNet that use $3 \times 3$ kernels and a stride of 2.

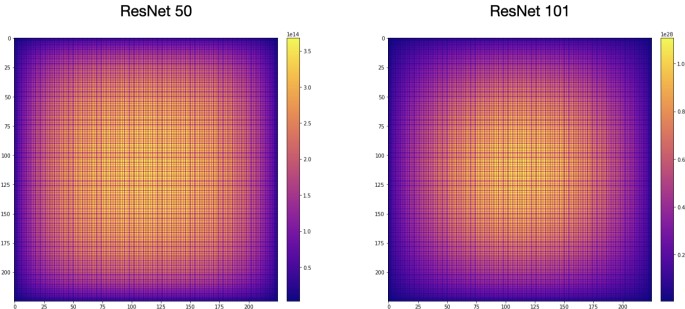

Figure 9: The foveation maps of two ResNet architectures under 0 padding, illustrated with a $225 \times 225$ input. Compared with ResNet-50, ResNet-101 has twice the number of convolutional layers with non-unitary filter sizes. Accordingly, the extent of the foveation effect is doubled.

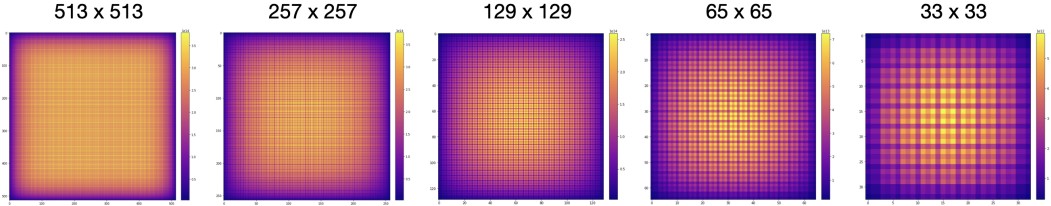

Figure 10: The foveation maps of ResNet-50 under 0 padding, illustrated with inputs of different size. The smaller the input, the larger the *relative* extent of foveation.

In the next figure, we illustrate how uneven application of padding impacts the foveation maps.
**Note:** It is possible to rectify the skewness in the 2nd foveation map by alternating the side where one-sided padding is applied between successive downsampling layers. This, however, does not mitigate the skewness in the learned filters (see next Section).

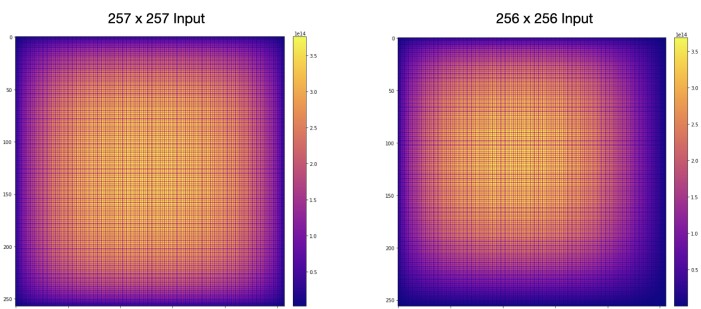

Figure 11: The foveation maps of ResNet-50 under 0 padding, illustrated with two input sizes. With a $257 \times 257$ input, the padding is evenly applied at all downsampling layers, leading to a symmetric foveation map. With a $256 \times 256$ input, the padding is applied only to the left and top sides of feature maps at all downsampling layers, which limits the number of convolutional input-output paths for pixels in the bottom and right sides as evident in the skewed foveation map.

## C    THE IMPACT OF THE PADDING METHOD ON LEARNED WEIGHTS

In the presence of uneven application of padding, 0-padding causes skewness in the learned weights because the filters are exposed more frequently to feature-map patches with zeros at their top and left sides. Redundancy methods such as circular or mirror padding mitigate such skewness because they fill the padding areas with values taken from the feature maps. PartialConv also mitigates such skewness because it assumes the pixels in the padding area are missing, and rescales the partial convolutional sum to account for them. Below we show the effectiveness of these alternatives in mitigating the skewness in three ResNet architectures.

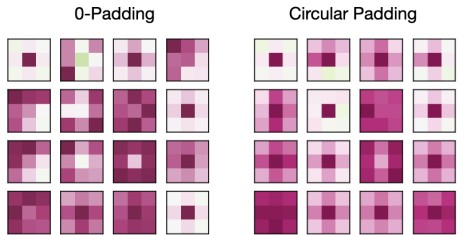 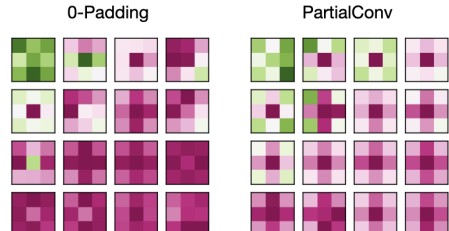

(a) Mean filters of ResNet-18 trained on 224×224 images under two padding methods, reaching 69.93% top-1 accuracy under 0-padding and 70.28% top-1 accuracy under circular padding.

(b) Mean filters of ResNet-50 trained on 224×224 images under two padding methods, reaching 76.15% top-1 accuracy under 0-padding and 76.61% top-1 accuracy under PartialConv.

Figure 12: Mean filters of two ResNet models trained on ImageNet with 224×224 images. The input size causes uneven application of padding, leading to frequent asymmetries in the mean filters under 0 padding. We illustrate how two alternatives, circular padding and PartialConv [23], enable learning highly-symmetric mean filters despite the uneven application of padding.

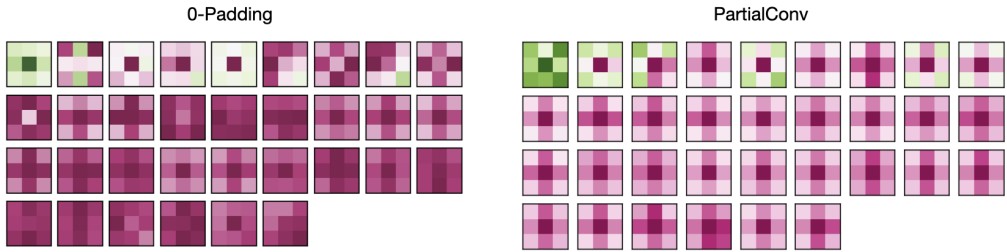

Figure 13: Mean filters of ResNet-101 trained on ImageNet with 224×224 images under both 0-padding and PartialConv [23]. The input size causes uneven application of padding, leading to frequent asymmetries in the mean filters under 0 padding. In contrast, PartialConv produces highly symmetric mean filters, thanks for its treatment of pixels outside the feature map as missing values.

**What if no padding is applied during downsampling?**    VGG models perform downsampling using $2 \times 2$ pooling layers that do not apply any padding. Accordingly, the mean filters do not exhibit significant skewness, even if the input size does not satisfy Eq 4:

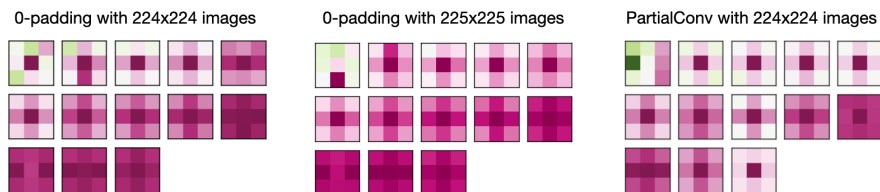

Figure 14: Mean filters of VGG-16 trained on ImageNet under different conditions. Most mean filters exhibit high symmetry when trained with 225×225 images where the size violates Eq. 4.

# D  THE IMPACT OF PADDING METHODS ON FEATURE-MAP ARTIFACTS

We show per-layer mean feature maps in ResNet-18 under different padding methods. The mean maps are averaged over 20 input samples generated at random.

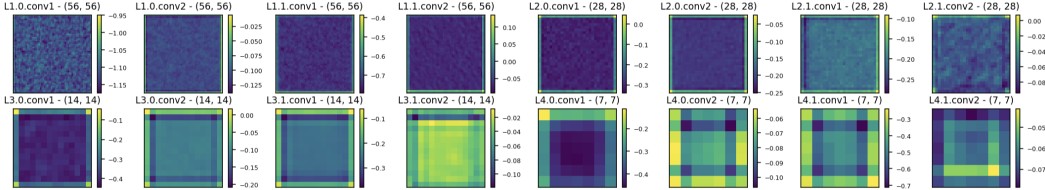

Figure 15: Feature map artifacts under zero padding. Line artifacts accumulate to become significant and asymmetric at deeper layers.

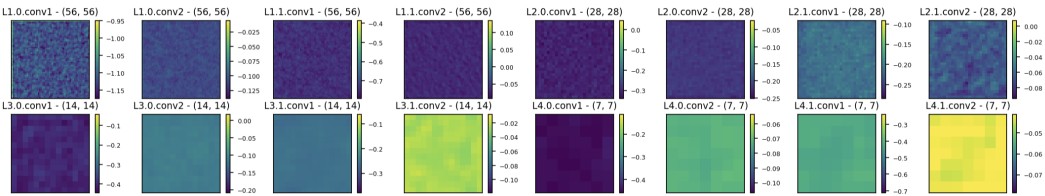

Figure 16: Circular padding largely preserves the randomness and mitigates line artifacts.

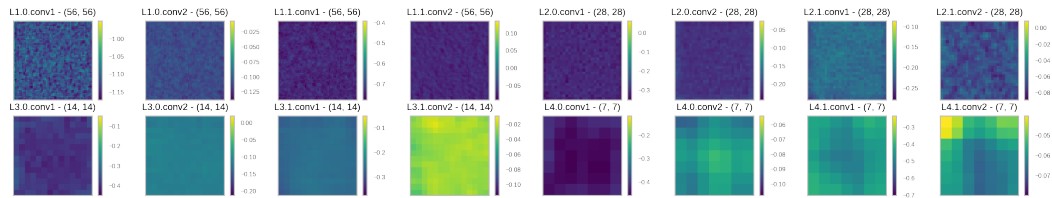

Figure 17: SYMMETRIC mirror padding also preserves the randomness and mitigates line artifacts.

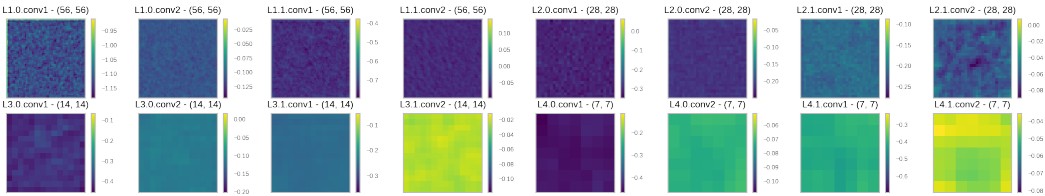

Figure 18: REFLECT mirror padding also preserves the randomness and mitigates line artifacts.

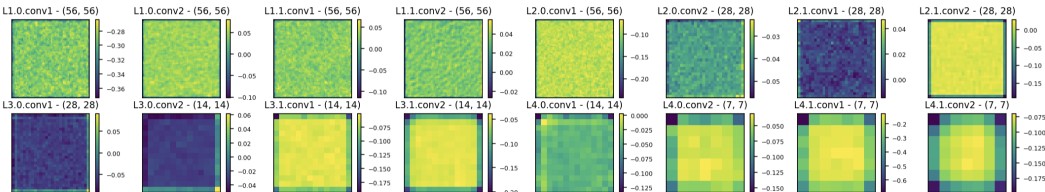

Figure 19: PartialConv [23] highly preserves the symmetry of the feature maps. The scaling factors it uses can break the randomness at the boundary.

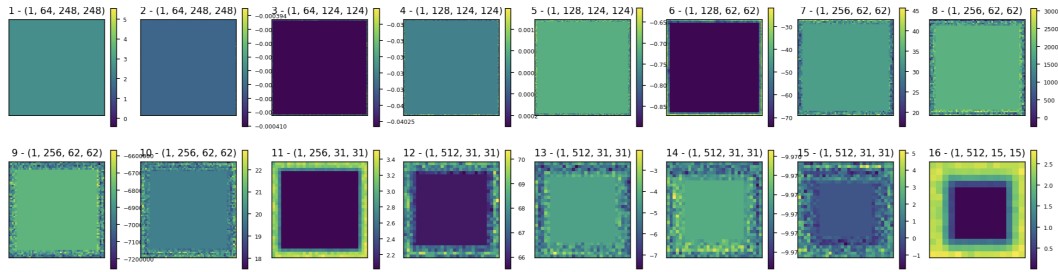

Figure 20: Feature map artifacts of a VGG-19 model under Distribution Padding (interpolation mode) [30]. Due to multiple resize operations used to fill the padding area, the artifacts grow from the boundary inwards. We use a saturated constant input to make the effect visible.

# E  THE IMPACT OF ANTIALIASING ON THE LEARNED WEIGHTS

We demonstrate how antialiasing [43] significantly reduces the asymmetry of mean filters around downsampling layers, even in the presence of unevenly-applied zero padding.

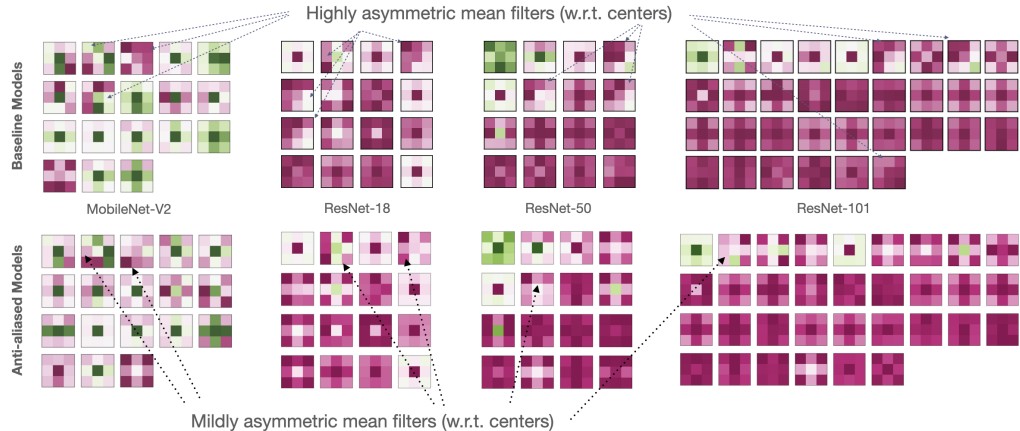

Figure 21: Mean filters of four models trained on ImageNet with 224×224 images under 0-padding both without and with antialiasing.

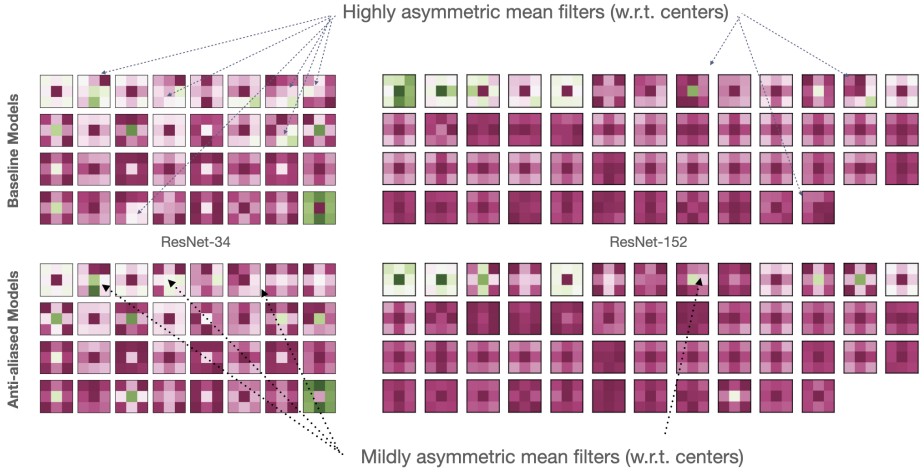

Figure 22: Mean filters of two models trained on ImageNet with 224×224 images under 0-padding both without and with antialiasing.

# F FOVEATION ANALYSIS OF PADDING ALGORITHMS

Refer to http://mind-the-pad.github.io for an interactive and animated visual illustration of padding algorithms and their foveation behavior. This appendix serves as a print version.

Among the SAME padding algorithms we discussed in the manuscript, two algorithms warrant that each input pixel is involved in an equal number of convolutional operations, leading to uniform foveation maps: circular padding and SYMMETRIC mirror padding. In contrast, this number varies under zero padding, REFLECT mirror padding, replication padding, and partial convolution.

We illustrate in detail how each padding algorithm treats the input pixels. For this purpose we illustrate step by step how each pixel is processed by the convolutional kernel. We choose a set of pixels that are sufficient to expose the behavior of the respective algorithm. This set spans an area within two or three pixels from the boundary that encompasses all relevant cases for the analysis and is situated at the top-left corner. The behavior at the other corners is analogous.

All illustrations use a stride of 1. Except for VALID, all configurations warrant SAME padding.

- **VALID Padding:** This algorithm is illustrated on a $3 \times 3$ kernel without dilation. A larger kernel size or dilation factor will increase the foveation effect.

- **Zero Padding:** This algorithm is illustrated on a $3 \times 3$ kernel without dilation. A larger kernel size or dilation factor will increase the foveation effect.

- **Circular Padding:** This algorithm is illustrated on a $3 \times 3$ kernel without dilation. It is straightforward to prove that the algorithm warrants equal treatment of the pixels irrespective of the kernel size or dilation factor. This is because it effectively applies circular convolution: Once the kernel hits one side, it can seamlessly operate on the pixels of the other side. Circular convolution hence renders the feature map as infinite to the kernel, warranting that edge pixels are treated in the same manner as interior pixels.

- **Mirror Padding (SYMMETRIC):** This algorithm warrants that each pixel is involved in the same number of convolutional operations. It is important to notice that, unlike under circular convolution, these operations do not utilize the kernel pixels uniformly as we demonstrate in detail. We illustrate the algorithm behavior under the following settings:
  - $3 \times 3$ kernel and dilation factor of 1.
  - $5 \times 5$ kernel and dilation factor of 1.
  - $3 \times 3$ kernel and dilation factor of 2.
  - $2 \times 2$ kernel and dilation factor of 1, along with a grouped padding strategy to compensate for uneven padding [41].
  - $4 \times 4$ kernel size and dilation factor of 1, along with a grouped padding strategy.

- **Mirror Padding (REFLECT):** This algorithm is illustrated on a $3 \times 3$ kernel without dilation.

- **Replication Padding:** This algorithm is illustrated on a $5 \times 5$ kernel without dilation. We choose this kernel size since a $3 \times 3$ kernel under SAME padding would render the algorithm equivalent to SYMMETRIC mirror padding.

- **Partial Convolution:** This algorithm is illustrated on a $3 \times 3$ kernel without dilation. Its foveation behavior is analogous to REFLECT mirror padding.

# VALID Padding   Illustrated on a 3x3 kernel

## Input

| a | b | c | .. | .. |
|---|---|---|----|----|
| d | e | f | .. | .. |
| g | h | i | .. | .. |
| .. | .. | .. | .. | .. |
| .. | .. | .. | .. | .. |

## # of conv ops each pixel is involved in

| 1 | 2 | 3 | 3 | 3 |
|---|---|---|---|---|
| 2 | 4 | 6 | 6 | 6 |
| 3 | 6 | **9** | 9 | 9 |
| 3 | 6 | 9 | 9 | 9 |
| 3 | 6 | 9 | 9 | 9 |

## Which kernel cells these ops utilize?

a: 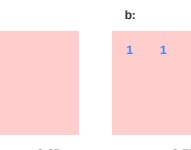  b: 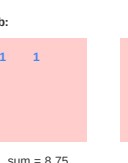  c: 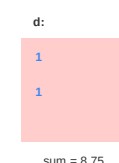  d: 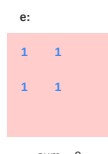  e: 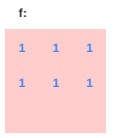  f: 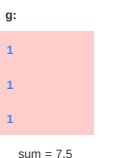  g: 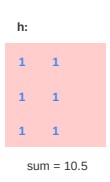  h: 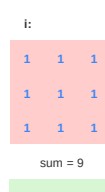  i:

sum = 6.25    sum = 8.75    sum = 7.5    sum = 8.75    sum = 9    sum = 10.5    sum = 7.5    sum = 10.5    sum = 9

uniform

## Detailed Illustration of how the counts are derived

Convolutions involving (a)

Convolutions involving (b)

Convolutions involving (c)

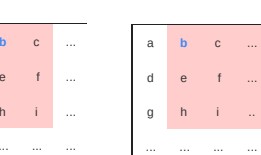

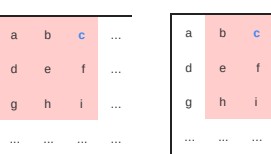

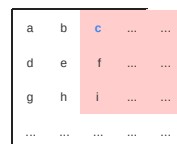

Convolutions involving (d): rotated version of (b)

Convolutions involving (e)

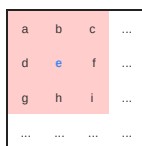 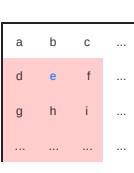 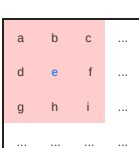 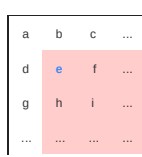

Convolutions involving (f)

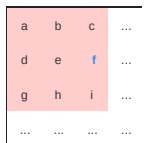 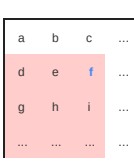 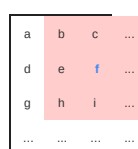 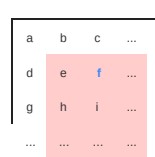 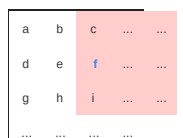 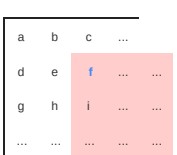

Convolutions involving (g): Rotated version of (c)

Convolutions involving (h): Rotated version of (f)

Convolutions involving (i): Regular uniform treatment

# Zero Padding

## Illustrated on 3x3 kernel and 1-pixel padding

**Original Input**

| | | | | |
|---|---|---|---|---|
| a | b | .. | .. | .. |
| c | d | .. | .. | .. |
| .. | .. | .. | .. | .. |
| .. | .. | .. | .. | .. |
| .. | .. | .. | .. | .. |

**Padded Input**

| 0 | 0 | 0 | .. | .. | .. |
|---|---|---|---|---|---|
| 0 | a | b | .. | .. | .. |
| 0 | c | d | .. | .. | .. |
| 0 | .. | .. | .. | .. | .. |
| .. | .. | .. | .. | .. | .. |
| .. | .. | .. | .. | .. | .. |

**# of conv ops each pixel is involved in**

| 4 | 6 | 6 | 6 | 6 |
|---|---|---|---|---|
| 6 | 9 | 9 | 9 | 9 |
| 6 | 9 | 9 | 9 | 9 |
| 6 | 9 | 9 | 9 | 9 |
| 6 | 9 | 9 | 9 | 9 |

## Which kernel cells these ops utilize?

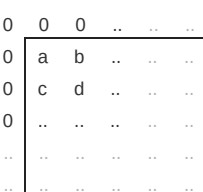

a — sum = 4  
b — sum = 6  
c — sum = 6  
d — sum = 9 (uniform)

## Detailed Illustration of how the counts are derived

Convolutions involving (a)

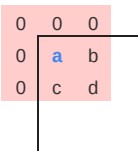 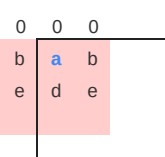 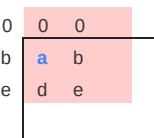

Convolutions involving (b)

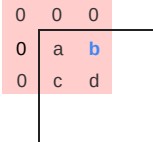 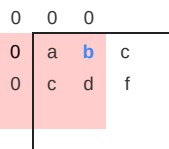 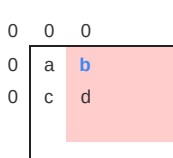

Other border cases are translation or rotation of (a) or (b)

# Circular Padding

## Illustrated on 3x3 kernel and 1-pixel padding

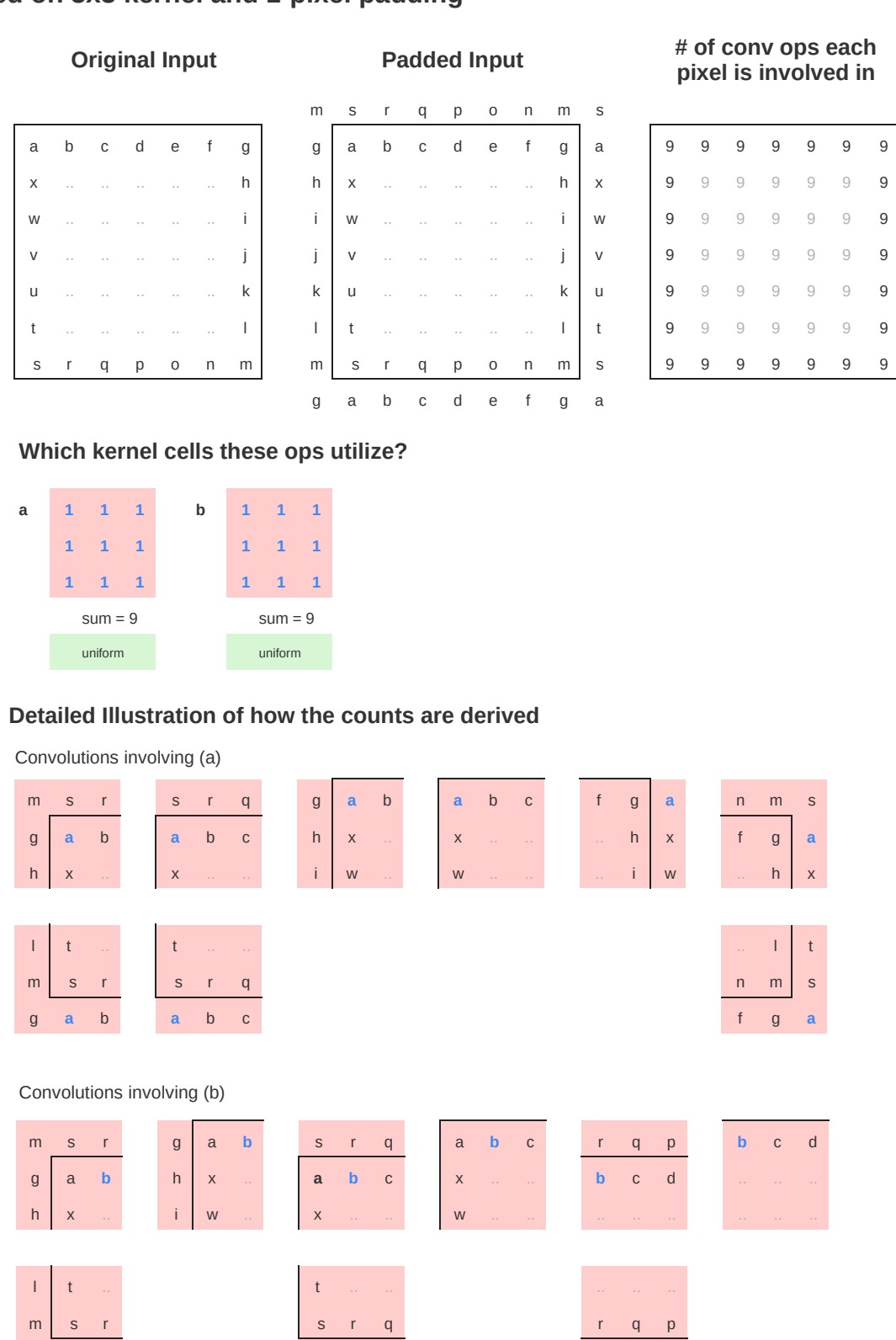

**Original Input**

**Padded Input**

**# of conv ops each pixel is involved in**

**Which kernel cells these ops utilize?**

**Detailed Illustration of how the counts are derived**

Convolutions involving (a)

Convolutions involving (b)

Other border cases are translation or rotation of (a) or (b)

# Mirror Padding (SYMMETRIC)

## Illustrated on 3x3 kernel and 1-pixel padding

**Original Input**

| a | b | c | .. | .. |
|---|---|---|----|----|
| d | e | f | .. | .. |
| g | h | i | .. | .. |
| .. | .. | .. | .. | .. |
| .. | .. | .. | .. | .. |

**Padded Input**

|   | a | a | b | c | .. | .. |
|---|---|---|---|---|----|----|
| a | a | b | c | .. | .. |   |
| d | d | e | f | .. | .. |   |
| g | g | h | i | .. | .. |   |
| .. | .. | .. | .. | .. | .. |   |
| .. | .. | .. | .. | .. | .. |   |

**# of conv ops each pixel is involved in**

| 9 | 9 | 9 | 9 | 9 |
|---|---|---|---|---|
| 9 | 9 | 9 | 9 | 9 |
| 9 | 9 | 9 | 9 | 9 |
| 9 | 9 | 9 | 9 | 9 |
| 9 | 9 | 9 | 9 | 9 |

## Which kernel cells these ops utilize?

**a:**
| 4 | 2 |
|---|---|
| 2 | 1 |

sum = 9

**b:**
| 2 | 2 | 2 |
|---|---|---|
| 1 | 1 | 1 |

sum = 9

**c:**
| 2 | 2 | 2 |
|---|---|---|
| 1 | 1 | 1 |

sum = 9

**d:**
| 2 | 1 |
|---|---|
| 2 | 1 |
| 2 | 1 |

sum = 9

**e:**
| 1 | 1 | 1 |
|---|---|---|
| 1 | 1 | 1 |
| 1 | 1 | 1 |

sum = 9
uniform

**f:**
| 1 | 1 | 1 |
|---|---|---|
| 1 | 1 | 1 |
| 1 | 1 | 1 |

sum = 9
uniform

**g:**
| 2 | 1 |
|---|---|
| 2 | 1 |
| 2 | 1 |

sum = 9

**h:**
| 1 | 1 | 1 |
|---|---|---|
| 1 | 1 | 1 |
| 1 | 1 | 1 |

sum = 9
uniform

**i:**
| 1 | 1 | 1 |
|---|---|---|
| 1 | 1 | 1 |
| 1 | 1 | 1 |

sum = 9
uniform

## Detailed Illustration of how the counts are derived

Convolutions involving (a)

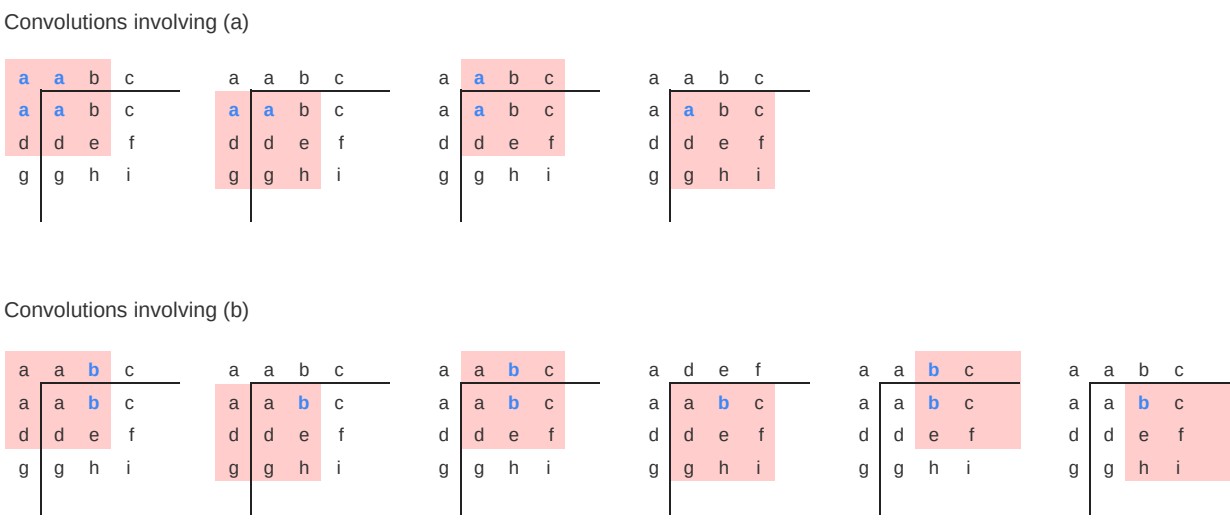

Convolutions involving (b)

Convolutions involving (c)

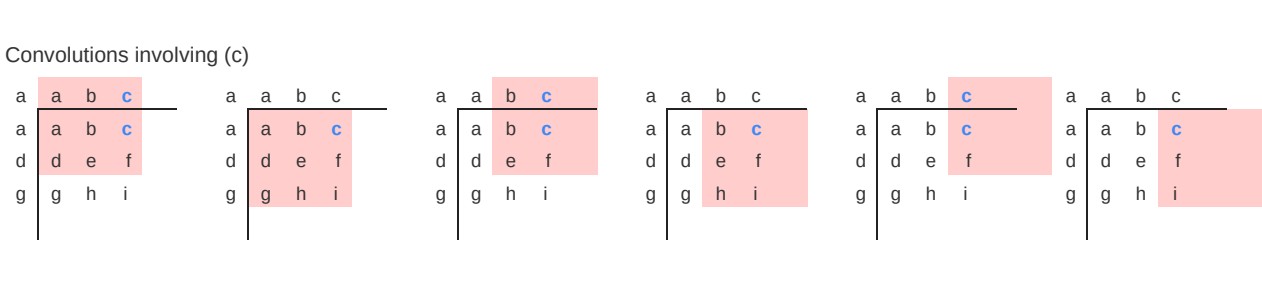

Convolutions involving (d): Rotated version of (b)

Convolutions involving (e)

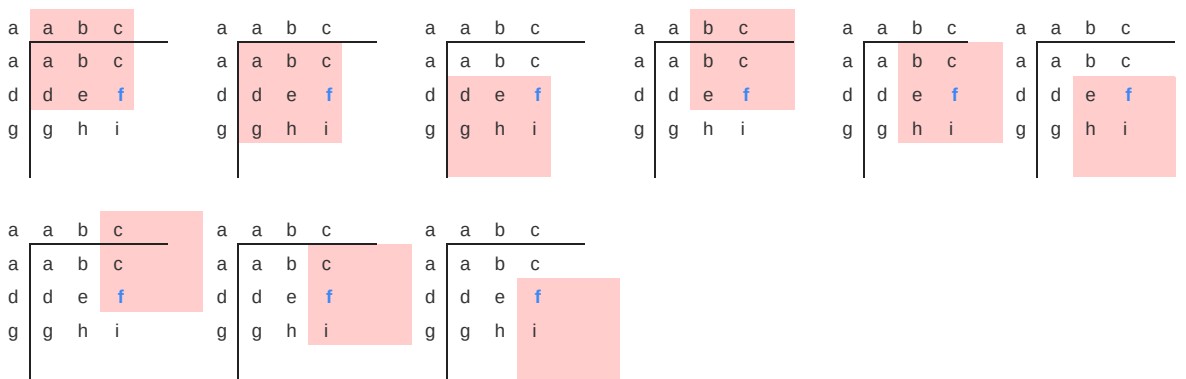

Convolutions involving (f)

Convolutions involving (g): Rotated version of (c)

Convolutions involving (h): Rotated version of (f)

Convolutions involving (i): Regular uniform treatment

## Mirror Padding (SYMMETRIC)

### Illustrated on 5x5 kernel and 2-pixel padding

**Original Input**

```
a  b  c  ..  ..
d  e  f  ..  ..
g  h  i  ..  ..
..  ..  ..  ..  ..
..  ..  ..  ..  ..
```

**Padded Input**

```
e  d  d  e  f  ..  ..
b  a  a  b  c  ..  ..
b  a  a  b  c  ..  ..
e  d  d  e  f  ..  ..
h  g  g  h  i  ..  ..
..  ..  ..  ..  ..  ..  ..
..  ..  ..  ..  ..  ..  ..
```

**# of conv ops for each pixel**

```
25  25  25  25  25
25  25  25  25  25
25  25  25  25  25
25  25  25  25  25
25  25  25  25  25
```

## Which kernel cells these ops utilize?

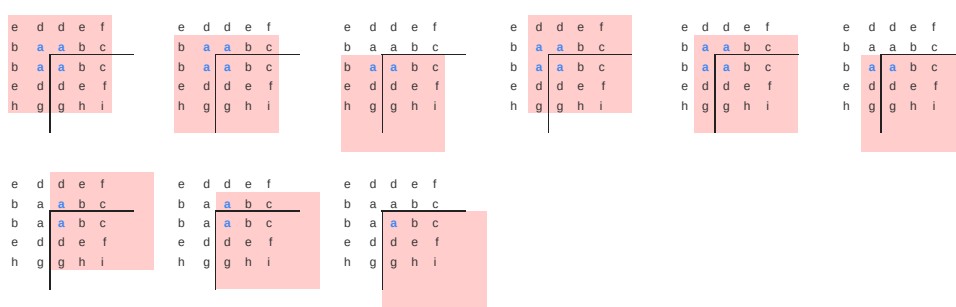

a: sum = 25
b: sum = 25
c: sum = 25
d: sum = 25
e: sum = 25
f: sum = 25
g: sum = 25
h: sum = 25
i: sum = 25 (uniform)

## Detailed Illustration of how the counts are derived

### Convolutions involving (a)

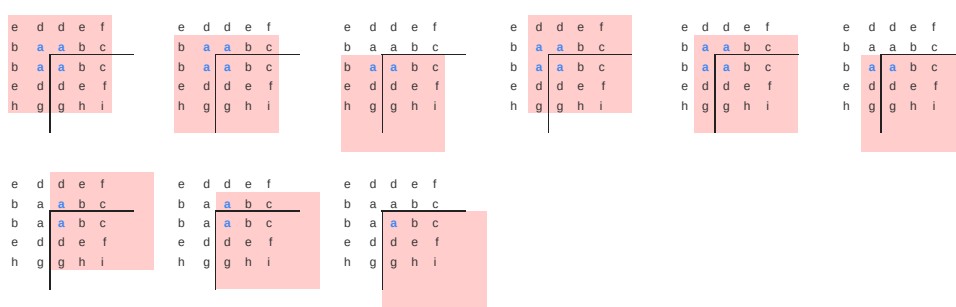

### Convolutions involving (b)

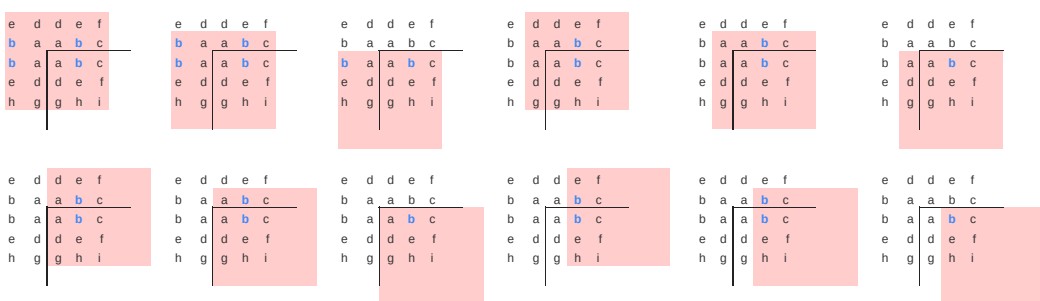

### Convolutions involving (c)

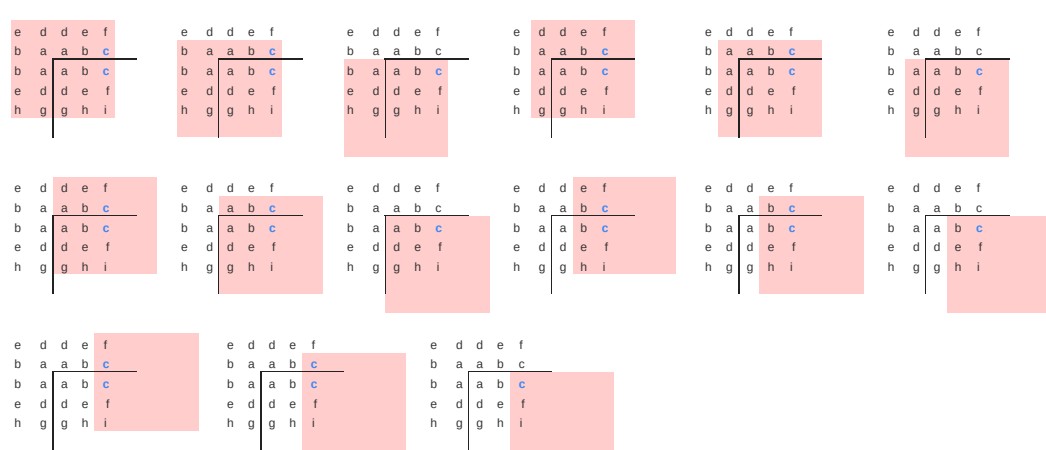

Convolutions involving (d): Rotated version of (b)

Convolutions involving (e)

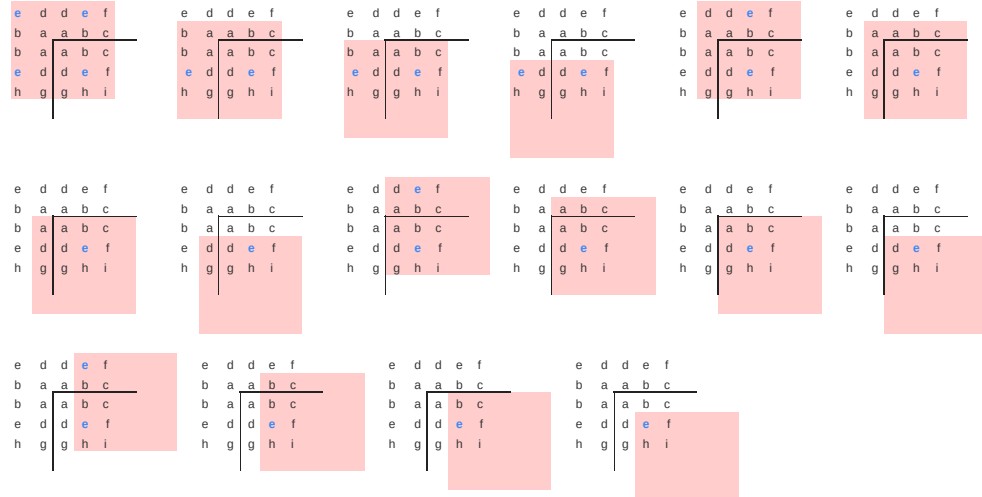

Convolutions involving (f)

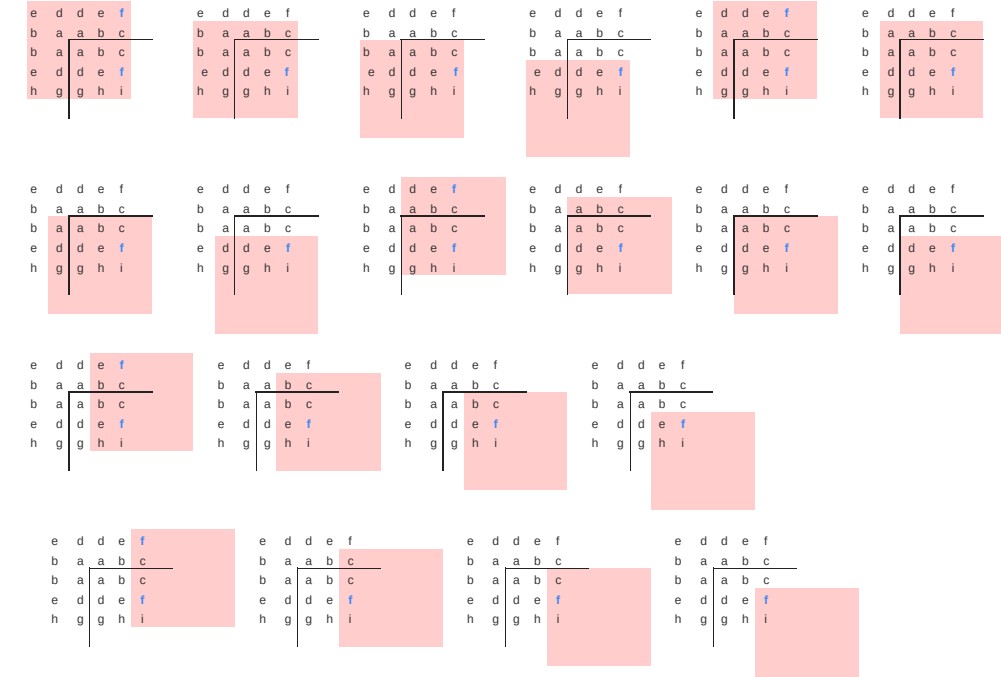

Convolutions involving (g): Rotated version of (c)

Convolutions involving (h): Rotated version of (f)

Convolutions involving (i): Regular uniform treatment

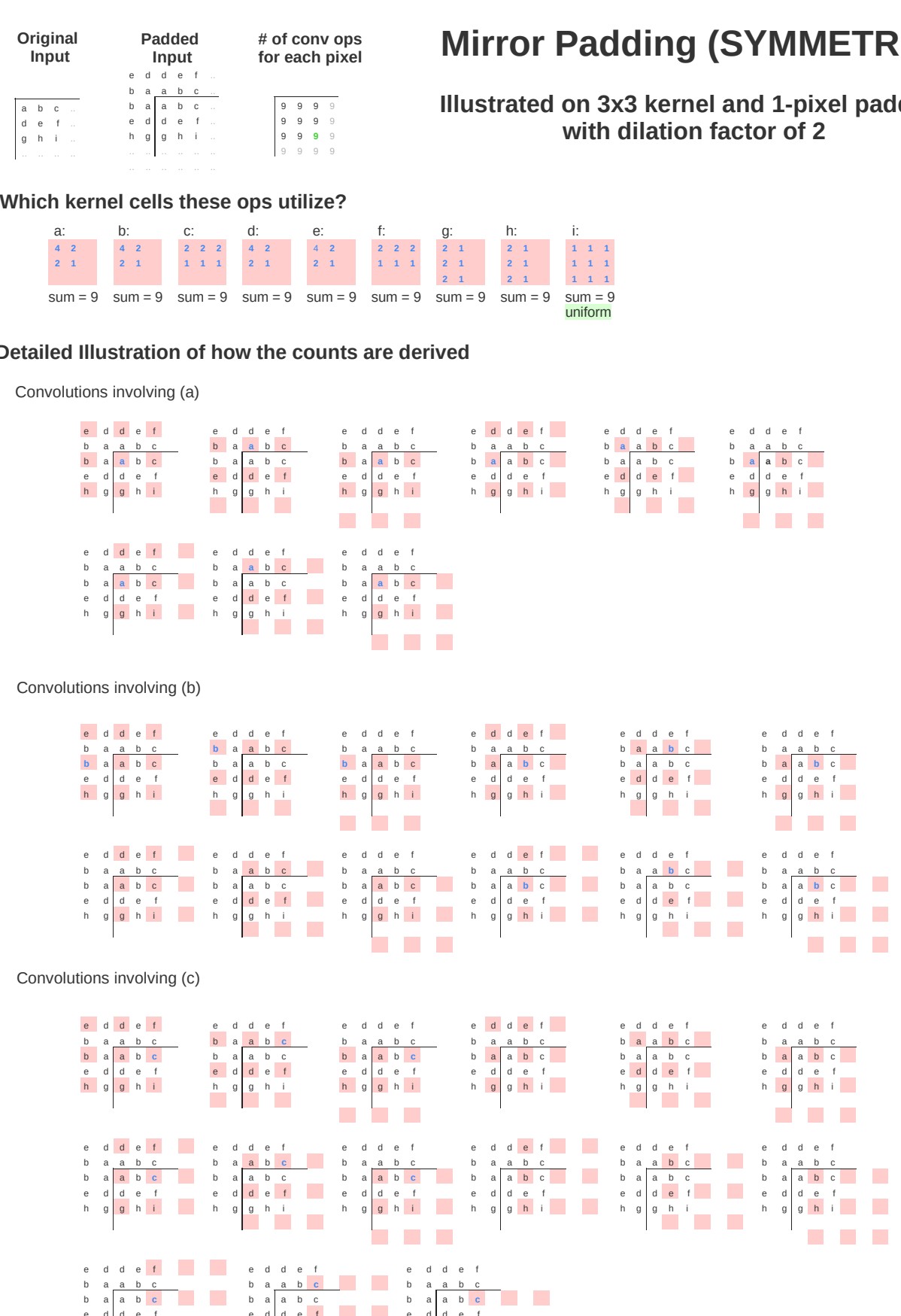

# Mirror Padding (SYMMETRIC)

## Illustrated on 3x3 kernel and 1-pixel padding with dilation factor of 2

**Original Input**

**Padded Input**

**# of conv ops for each pixel**

**Which kernel cells these ops utilize?**

a: sum = 9
b: sum = 9
c: sum = 9
d: sum = 9
e: sum = 9
f: sum = 9
g: sum = 9
h: sum = 9
i: sum = 9
uniform

**Detailed Illustration of how the counts are derived**

Convolutions involving (a)

Convolutions involving (b)

Convolutions involving (c)

Convolutions involving (d): Rotated version of (b)

Convolutions involving (e)

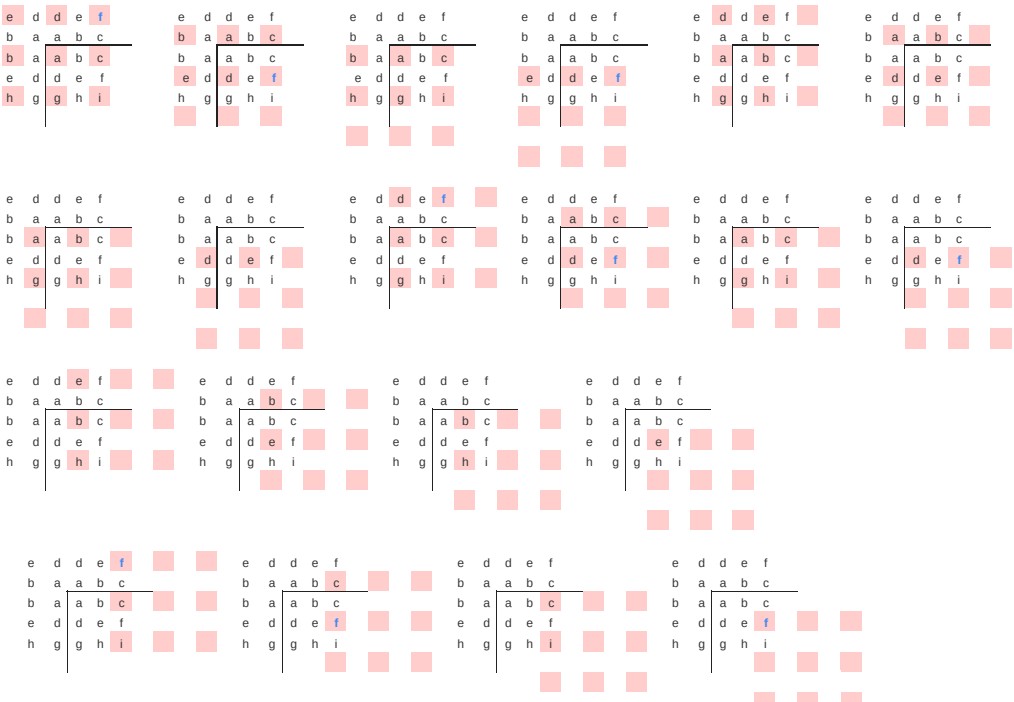

Convolutions involving (f)

Convolutions involving (g): Rotated version of (c)

Convolutions involving (h): Rotated version of (f)

Convolutions involving (i): Regular uniform treatment

# Mirror Padding (SYMMETRIC) with Grouping

**Illustrated on 2x2 kernel and 1-pixel padding**
**A grouped padding strategy is applied to balance uneven padding (Wu et al 2019)**

| Original Input | Padded at top-left corner | Padded at bottom-left corner | Padded at top-right corner | Padded at bottom-right corner |
|---|---|---|---|---|

Original Input:
| a | b | .. |
| c | d | .. |
| | | |

Padded at top-left corner:
| a | a | b | |
| a | a | b | .. |
| c | c | d | .. |
| | .. | .. | .. |

Padded at bottom-left corner:
| a | a | b | .. |
| c | c | d | .. |
| .. | .. | .. | |

Padded at top-right corner:
| a | b | |
| a | b | .. |
| c | d | .. |
| .. | .. | .. |

Padded at bottom-right corner:
| a | b | .. |
| c | d | .. |
| .. | .. | .. |

## Number of conv ops each pixel is involved in

| Padded at top-left corner | Padded at bottom-left corner | Padded at top-right corner | Padded at bottom-right corner | Average (grouped padding strategy) |
|---|---|---|---|---|

Padded at top-left corner:
| 9 | 6 | 6 |
| 6 | 4 | 4 |
| 6 | 4 | 4 |

Padded at bottom-left corner:
| 3 | 2 | 2 |
| 6 | 4 | 4 |
| 6 | 4 | 4 |

Padded at top-right corner:
| 3 | 6 | 6 |
| 2 | 4 | 4 |
| 2 | 4 | 4 |

Padded at bottom-right corner:
| 1 | 2 | 2 |
| 2 | 4 | 4 |
| 2 | 4 | 4 |

Average (grouped padding strategy):
| 4 | 4 | 4 |
| 4 | **4** | 4 |
| 4 | 4 | 4 |

## Which kernel cells these ops utilize?

| | Padded at top-left corner | Padded at bottom-left corner | Padded at top-right corner | Padded at bottom-right corner | Average (grouped padding strategy) |
|---|---|---|---|---|---|

**Convolutions involving (a)**

top-left:
| 3 | 2 |
| 2 | 2 |
sum = 9

bottom-left:
| 2 | 1 |
sum = 3

top-right:
| 2 |
| 1 |
sum = 3

bottom-right:
| 1 |
sum = 1

Average:
| 2 | 0.75 |
| 0.75 | 0.5 |
sum = 4

**Convolutions involving (b)**

top-left:
| 2 | 2 |
| 1 | 1 |
sum = 6

bottom-left:
| 1 | 1 |
sum = 2

top-right:
| 2 | 2 |
| 1 | 1 |
sum = 6

bottom-right:
| 1 | 1 |
sum = 2

Average:
| 1.5 | 1.5 |
| 0.5 | 0.5 |
sum = 4

**Convolutions involving (c)**

top-left:
| 2 | 1 |
| 2 | 1 |
sum = 6

bottom-left:
| 2 | 1 |
| 2 | 1 |
sum = 6

top-right:
| 1 |
| 1 |
sum = 2

bottom-right:
| 1 |
| 1 |
sum = 2

Average:
| 1.5 | 0.5 |
| 1.5 | 0.5 |
sum = 4

**Convolutions involving (d)**

top-left:
| 1 | 1 |
| 1 | 1 |
sum = 4
uniform

bottom-left:
| 1 | 1 |
| 1 | 1 |
sum = 4
uniform

top-right:
| 1 | 1 |
| 1 | 1 |
sum = 4
uniform

bottom-right:
| 1 | 1 |
| 1 | 1 |
sum = 4
uniform

Average:
| 1 | 1 |
| 1 | 1 |
sum = 4
uniform

# Detailed Illustration of how the counts are derived

Convolutions involving (a)

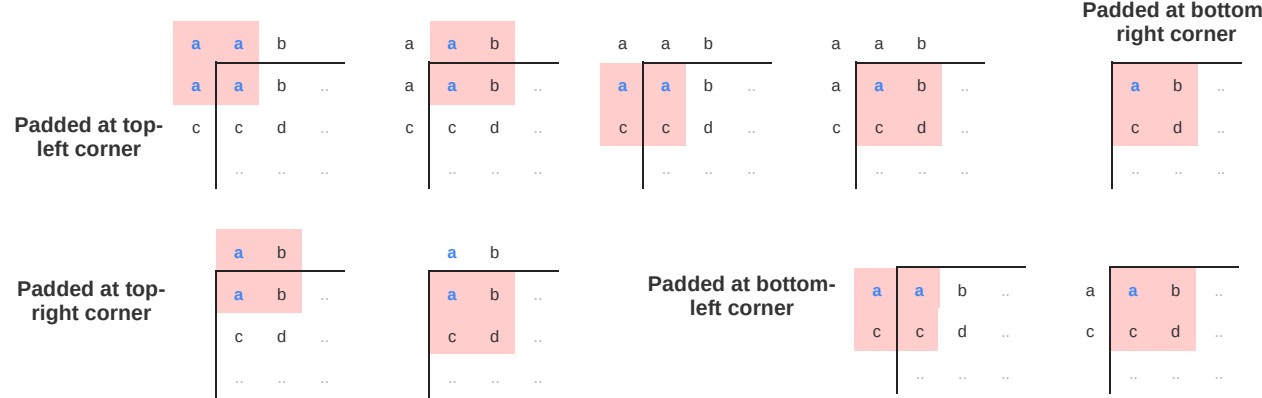

Convolutions involving (b)

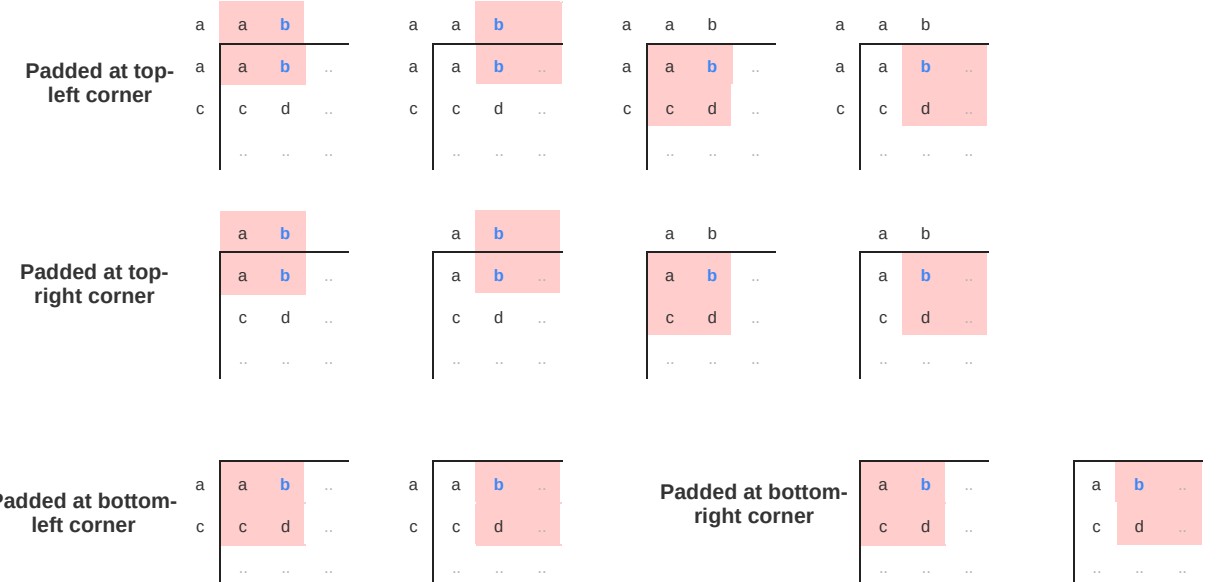

Convolutions involving (c): Rotated version of (b)

Convolutions involving (d): Rotated version of (a)

# Mirror Padding (SYMMETRIC) with Grouping
## Illustrated on 4x4 kernel and 1-pixel padding

### Original Input

|   |   |   |   |
|---|---|---|---|
| a | b | c | .. |
| d | e | f | .. |
| g | h | i |   |
| .. | .. | .. | .. |

### Padded at top-left

|   |   |   |   |   |
|---|---|---|---|---|
| e | d | d | e | f |
| b | a | a | b | c |
| b | a | a | b | c | .. |
| e | d | d | e | f | .. |
| h | g | g | h | i |
|   | .. | .. | .. | .. |

### Padded at bottom-left corner

|   |   |   |   |   |
|---|---|---|---|---|
| b | a | a | b | c |
| b | a | a | b | c | .. |
| e | d | d | e | f | .. |
| h | g | g | h | i |
| .. | .. | .. | .. |

### Padded at top-right

|   |   |   |   |
|---|---|---|---|
| d | d | e | f |
| a | a | b | c |
| a | a | b | c | .. |
| d | d | e | f | .. |
| g | g | h | i |
| .. | .. | .. | .. |

### Padded at bottom-right corner

|   |   |   |   |
|---|---|---|---|
| a | a | b | c |
| a | a | b | c | .. |
| d | d | e | f | .. |
| g | g | h | i |
| .. | .. | .. | .. |

### Number of conv ops each pixel is involved in

**Padded at top-left**

| 25 | 25 | 20 | 20 | 20 |
|----|----|----|----|----|
| 25 | 25 | 20 | 20 | 20 |
| 20 | 20 | 16 | 16 | 16 |
| 20 | 20 | 16 | 16 | 16 |
| 20 | 20 | 16 | 16 | 16 |

**Padded at bottom-left**

| 15 | 15 | 12 | 12 | 12 |
|----|----|----|----|----|
| 15 | 15 | 12 | 12 | 12 |
| 20 | 20 | 16 | 16 | 16 |
| 20 | 20 | 16 | 16 | 16 |
| 20 | 20 | 16 | 16 | 16 |

**Padded at top-right**

| 15 | 15 | 20 | 20 | 20 |
|----|----|----|----|----|
| 15 | 15 | 20 | 20 | 20 |
| 12 | 12 | 16 | 16 | 16 |
| 12 | 12 | 16 | 16 | 16 |
| 12 | 12 | 16 | 16 | 16 |

**Padded at bottom-right**

| 9 | 9 | 12 | 12 | 12 |
|---|---|----|----|----|
| 9 | 9 | 12 | 12 | 12 |
| 12 | 12 | 16 | 16 | 16 |
| 12 | 12 | 16 | 16 | 16 |
| 12 | 12 | 16 | 16 | 16 |

**Average (grouped padding strategy)**

| 16 | 16 | 16 | 16 | 16 |
|----|----|----|----|----|
| 16 | 16 | 16 | 16 | 16 |
| 16 | 16 | 16 | 16 | 16 |
| 16 | 16 | 16 | 16 | 16 |
| 16 | 16 | 16 | 16 | 16 |

### Which kernel cells these ops utilize?

Columns: Padded at top-left | Padded at bottom-left | Padded at top-right | Padded at bottom-right | Average (grouped padding strategy)

**(a)**

top-left: 
| 4 | 4 | 2 |
|---|---|---|
| 4 | 4 | 2 |
| 2 | 2 | 1 |
sum = 25

bottom-left:
| 4 | 4 | 2 |
|---|---|---|
| 2 | 2 | 1 |
sum = 15

top-right:
| 4 | 2 |
|---|---|
| 4 | 2 |
| 2 | 1 |
sum = 15

bottom-right:
| 4 | 2 |
|---|---|
| 2 | 1 |
sum = 9

Average:
| 4 | 3 | 1 |
|---|---|---|
| 3 | 2.25 | 0.75 |
| 1 | 0.75 | 0.25 |
sum = 16

**(b)**

top-left:
| 4 | 2 | 2 | 2 |
|---|---|---|---|
| 4 | 2 | 2 | 2 |
| 2 | 1 | 1 | 1 |
sum = 25

bottom-left:
| 4 | 2 | 2 | 2 |
|---|---|---|---|
| 2 | 1 | 1 | 1 |
sum = 15

top-right:
| 2 | 2 | 2 |
|---|---|---|
| 2 | 2 | 2 |
| 1 | 1 | 1 |
sum = 15

bottom-right:
| 2 | 2 | 2 |
|---|---|---|
| 1 | 1 | 1 |
sum = 9

Average:
| 3 | 2 | 2 | 1 |
|---|---|---|---|
| 2.25 | 1.5 | 1.5 | 0.75 |
| 0.75 | 0.5 | 0.5 | 0.25 |
sum = 16

**(c)**

top-left:
| 2 | 2 | 2 | 2 |
|---|---|---|---|
| 2 | 2 | 2 | 2 |
| 1 | 1 | 1 | 1 |
sum = 20

bottom-left:
| 2 | 2 | 2 | 2 |
|---|---|---|---|
| 1 | 1 | 1 | 1 |
sum = 12

top-right:
| 2 | 2 | 2 | 2 |
|---|---|---|---|
| 2 | 2 | 2 | 2 |
| 1 | 1 | 1 | 1 |
sum = 20

bottom-right:
| 2 | 2 | 2 | 2 |
|---|---|---|---|
| 1 | 1 | 1 | 1 |
sum = 12

Average:
| 2 | 2 | 2 | 2 |
|---|---|---|---|
| 1.5 | 1.5 | 1.5 | 1.5 |
| 0.5 | 0.5 | 0.5 | 0.5 |
sum = 16

**(d)** rotated version of (b)

**(e)**

top-left:
| 4 | 2 | 2 | 2 |
|---|---|---|---|
| 2 | 1 | 1 | 1 |
| 2 | 1 | 1 | 1 |
| 2 | 1 | 1 | 1 |
sum = 25

bottom-left:
| 2 | 1 | 1 | 1 |
|---|---|---|---|
| 2 | 1 | 1 | 1 |
| 2 | 1 | 1 | 1 |
sum = 15

top-right:
| 2 | 2 | 2 |
|---|---|---|
| 1 | 1 | 1 |
| 1 | 1 | 1 |
| 1 | 1 | 1 |
sum = 15

bottom-right:
| 1 | 1 | 1 |
|---|---|---|
| 1 | 1 | 1 |
| 1 | 1 | 1 |
sum = 9

Average:
| 2.25 | 1.5 | 1.5 | 0.75 |
|------|-----|-----|------|
| 1.5 | 1 | 1 | 0.5 |
| 1.5 | 1 | 1 | 0.5 |
| 0.75 | 0.5 | 0.5 | 0.25 |
sum = 16

**(f)**

top-left:
| 2 | 2 | 2 | 2 |
|---|---|---|---|
| 1 | 1 | 1 | 1 |
| 1 | 1 | 1 | 1 |
| 1 | 1 | 1 | 1 |
sum = 20

bottom-left:
| 1 | 1 | 1 | 1 |
|---|---|---|---|
| 1 | 1 | 1 | 1 |
| 1 | 1 | 1 | 1 |
| 1 | 1 | 1 | 1 |
sum = 12

top-right:
| 2 | 2 | 2 | 2 |
|---|---|---|---|
| 1 | 1 | 1 | 1 |
| 1 | 1 | 1 | 1 |
| 1 | 1 | 1 | 1 |
sum = 20

bottom-right:
| 1 | 1 | 1 | 1 |
|---|---|---|---|
| 1 | 1 | 1 | 1 |
| 1 | 1 | 1 | 1 |
| 1 | 1 | 1 | 1 |
sum = 12

Average:
| 1.5 | 1.5 | 1.5 | 1.5 |
|-----|-----|-----|-----|
| 1 | 1 | 1 | 1 |
| 1 | 1 | 1 | 1 |
| 0.5 | 0.5 | 0.5 | 0.5 |
sum = 16

**(g)** rotated version of (c)

**(h)** rotated version of (f)

**(i)** regular treatment

# Replication Padding

## Illustrated on 5x5 kernel and 2-pixel padding

**Original Input**

| a | b | c | .. | .. |
| d | e | f | .. | .. |
| g | h | i | .. | .. |
| .. | .. | .. | .. | .. |
| .. | .. | .. | .. | .. |

**Padded Input**

| a | a | a | b | c | .. | .. |
| a | a | a | b | c | .. | .. |
| a | a | a | b | c | .. | .. |
| d | d | d | e | f | .. | .. |
| g | g | g | h | i | .. | .. |
| .. | .. | .. | .. | .. | .. | .. |
| .. | .. | .. | .. | .. | .. | .. |

**# of conv ops for each pixel**

| 36 | 24 | 30 | 30 | 30 |
| 24 | 16 | 20 | 20 | 20 |
| 30 | 20 | 25 | 25 | 25 |
| 30 | 20 | 25 | 25 | 25 |
| 30 | 20 | 25 | 25 | 25 |

## Which kernel cells these ops utilize?

a:

| 9 | 6 | 3 |
| 6 | 4 | 2 |
| 3 | 2 | 1 |

sum = 36

b:

| 3 | 3 | 3 | 3 |
| 2 | 2 | 2 | 2 |
| 1 | 1 | 1 | 1 |

sum = 24

c:

| 3 | 3 | 3 | 3 | 3 |
| 2 | 2 | 2 | 2 | 2 |
| 1 | 1 | 1 | 1 | 1 |

sum = 30

d:

| 3 | 2 | 1 |
| 3 | 2 | 1 |
| 3 | 2 | 1 |
| 3 | 2 | 1 |

sum = 24

e:

| 1 | 1 | 1 | 1 |
| 1 | 1 | 1 | 1 |
| 1 | 1 | 1 | 1 |
| 1 | 1 | 1 | 1 |

sum = 16

f:

| 1 | 1 | 1 | 1 | 1 |
| 1 | 1 | 1 | 1 | 1 |
| 1 | 1 | 1 | 1 | 1 |
| 1 | 1 | 1 | 1 | 1 |

sum = 20

g:

| 3 | 2 | 1 |
| 3 | 2 | 1 |
| 3 | 2 | 1 |
| 3 | 2 | 1 |
| 3 | 2 | 1 |

sum = 30

h:

| 1 | 1 | 1 | 1 |
| 1 | 1 | 1 | 1 |
| 1 | 1 | 1 | 1 |
| 1 | 1 | 1 | 1 |
| 1 | 1 | 1 | 1 |

sum = 20

i:

| 1 | 1 | 1 | 1 | 1 |
| 1 | 1 | 1 | 1 | 1 |
| 1 | 1 | 1 | 1 | 1 |
| 1 | 1 | 1 | 1 | 1 |
| 1 | 1 | 1 | 1 | 1 |

sum = 25
uniform

## Detailed Illustration of how the counts are derived

Convolutions involving (a)

Convolutions involving (b)

Convolutions involving (c)

Convolutions involving (d): Rotated version of (b)

Convolutions involving (e)

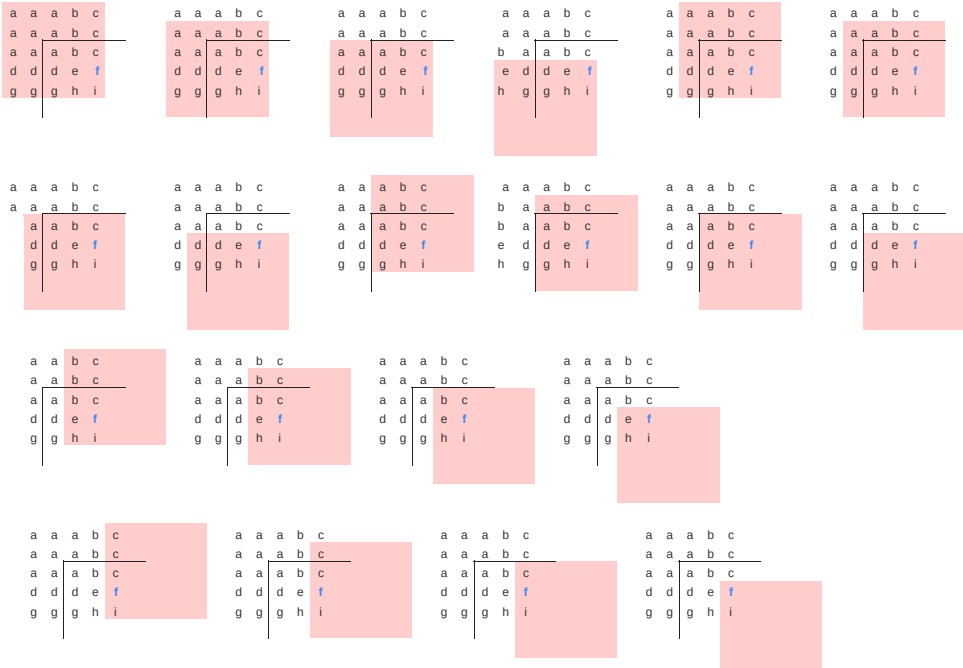

Convolutions involving (f)

Convolutions involving (g): Rotated version of (c)

Convolutions involving (h): Rotated version of (f)

Convolutions involving (i): Regular uniform treatment

# Mirror Padding (REFLECT)

## Illustrated on 3x3 kernel and 1-pixel padding

**Original Input**

| a | b | c | .. | .. |
|---|---|---|----|----|
| d | e | f | .. | .. |
| g | h | i | .. | .. |
| .. | .. | .. | .. | .. |
| .. | .. | .. | .. | .. |

**Padded Input**

|   | e | d | e | f | .. | .. |
|---|---|---|---|---|----|----|
| b | a | b | c | .. | .. |
| e | d | e | f | .. | .. |
| h | g | h | i | .. | .. |
| .. | .. | .. | .. | .. | .. |

**# of conv ops each pixel is involved in**

| 4 | 8 | 6 | 6 | 6 |
|---|---|---|---|---|
| 8 | 16 | 12 | 12 | 12 |
| 6 | 12 | 9 | 9 | 9 |
| 6 | 12 | 9 | 9 | 9 |
| 6 | 12 | 9 | 9 | 9 |

### Which kernel cells these ops utilize?

**a:**
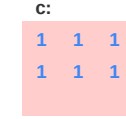

| 1 | 1 |
|---|---|
| 1 | 1 |

sum = 4

**b:**

| 2 | 1 | 1 |
|---|---|---|
| 2 | 1 | 1 |

sum = 8

**c:**

| 1 | 1 | 1 |
|---|---|---|
| 1 | 1 | 1 |

sum = 6

**d:**

| 2 | 2 |   |
|---|---|---|
| 1 | 1 |   |
| 1 | 1 |   |

sum = 8

**e:**

| 2 | 2 | 2 |
|---|---|---|
| 1 | 1 | 1 |
| 1 | 1 | 1 |

sum = 12

**f:**

| 4 | 2 | 2 |
|---|---|---|
| 2 | 1 | 1 |
| 2 | 1 | 1 |

sum = 16

**g:**

| 1 | 1 | 1 |
|---|---|---|
| 1 | 1 | 1 |

sum = 6

**h:**

| 2 | 1 | 1 |
|---|---|---|
| 2 | 1 | 1 |
| 2 | 1 | 1 |

sum = 12

**i:**

| 1 | 1 | 1 |
|---|---|---|
| 1 | 1 | 1 |
| 1 | 1 | 1 |

sum = 9

uniform

## Detailed Illustration of how the counts are derived

Convolutions involving (a)

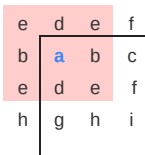

Convolutions involving (b)

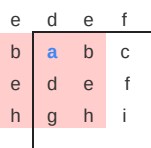
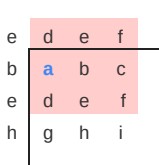
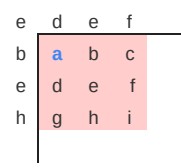

Convolutions involving (c)

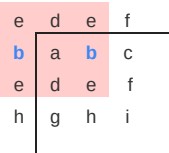
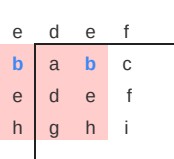
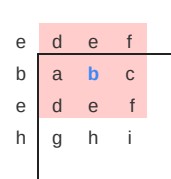
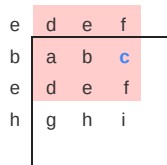
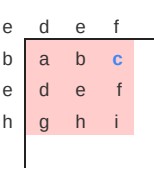
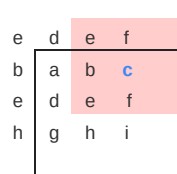

Convolutions involving (d): Rotated version of (b)

Convolutions involving (e)

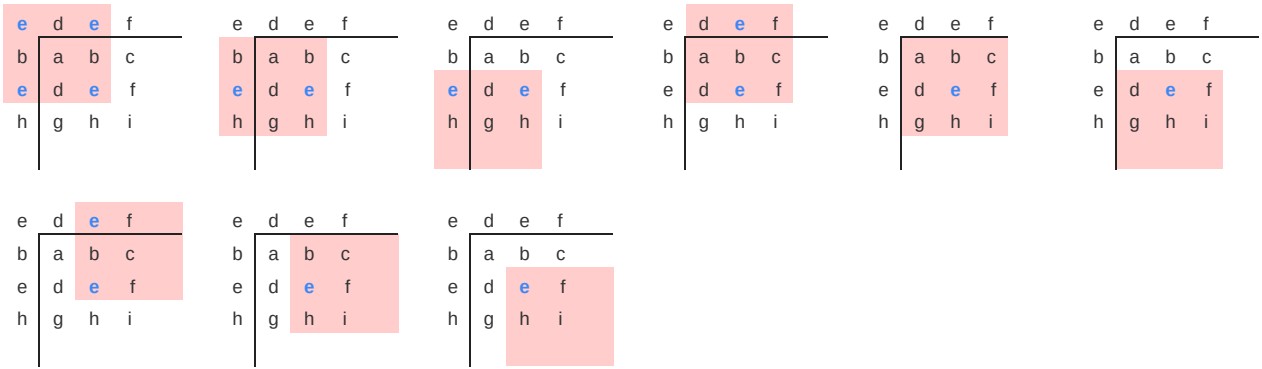

Convolutions involving (f)

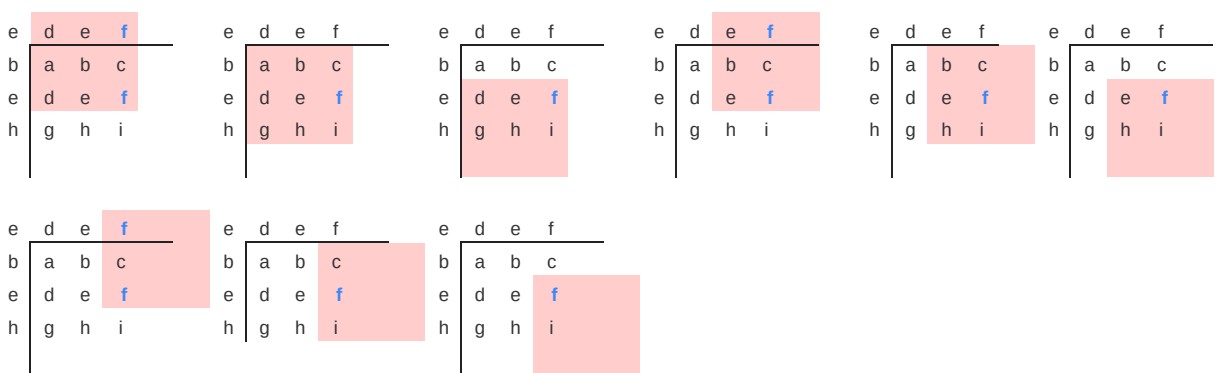

Convolutions involving (g): Rotated version of (c)

Convolutions involving (h): Rotated version of (f)

Convolutions involving (i): Regular uniform treatment

# Partial Convolution   Illustrated on a 3x3 kernel

**Input**

| | | | | |
|---|---|---|---|---|
| a | b | c | .. | .. |
| d | e | f | .. | .. |
| g | h | i | .. | .. |
| .. | .. | .. | .. | .. |
| .. | .. | .. | .. | .. |

**Weighted # of conv ops each pixel is involved in**

| | | | | |
|---|---|---|---|---|
| 6.25 | 8.75 | 7.5 | 7.5 | 7.5 |
| 8.75 | 12.25 | 10.5 | 10.5 | 10.5 |
| 7.5 | 10.5 | **9** | 9 | 9 |
| 7.5 | 10.5 | 9 | 9 | 9 |
| 7.5 | 10.5 | 9 | 9 | 9 |

## Which kernel cells these ops utilize?

a:
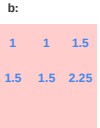
sum = 6.25

b:
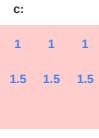
sum = 8.75

c:
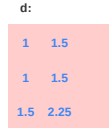
sum = 7.5

d:
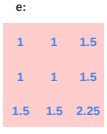
sum = 8.75

e:
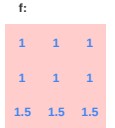
sum = 9

f:
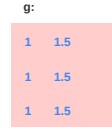
sum = 10.5

g:
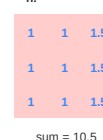
sum = 7.5

h:
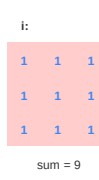
sum = 10.5

i:
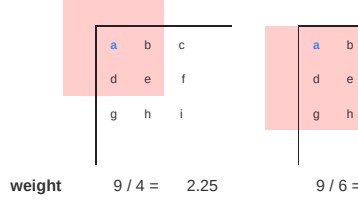
sum = 9

uniform

## Detailed Illustration of how the counts are derived

### Convolutions involving (a)

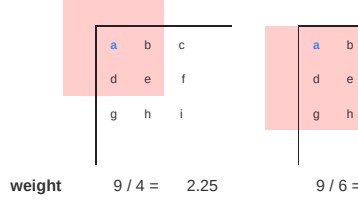

**weight**   9 / 4 = 2.25     9 / 6 = 1.5     9 / 6 = 1.5     9 / 9 = 1

**sum = 6.25**

### Convolutions involving (b)

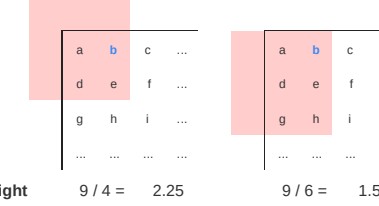

**weight**   9 / 4 = 2.25     9 / 6 = 1.5     9 / 6 = 1.5     9 / 9 = 1     9 / 6 = 1.5     9 / 9 = 1

**sum = 8.75**

### Convolutions involving (c)

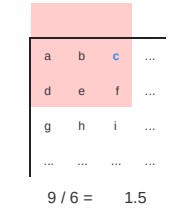

**weight**   9 / 6 = 1.5     9 / 9 = 1     9 / 6 = 1.5     9 / 9 = 1     9 / 6 = 1.5     9 / 9 = 1

**sum = 7.5**

Convolutions involving (d): rotated version of (b)

Convolutions involving (e)

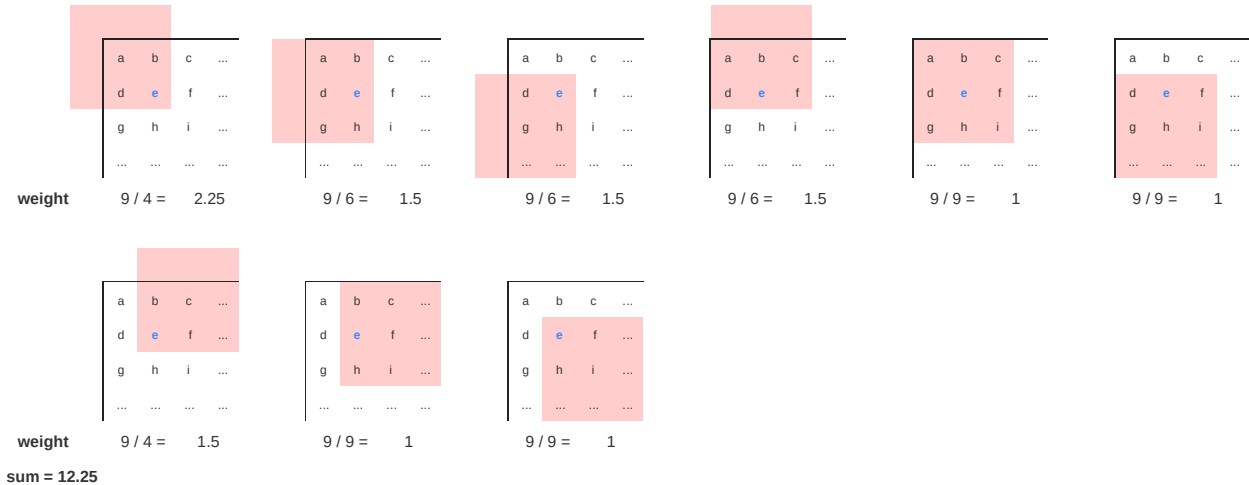

weight    9 / 4 =    2.25      9 / 6 =    1.5      9 / 6 =    1.5      9 / 6 =    1.5      9 / 9 =    1      9 / 9 =    1

weight    9 / 4 =    1.5      9 / 9 =    1      9 / 9 =    1

sum = 12.25

Convolutions involving (f)

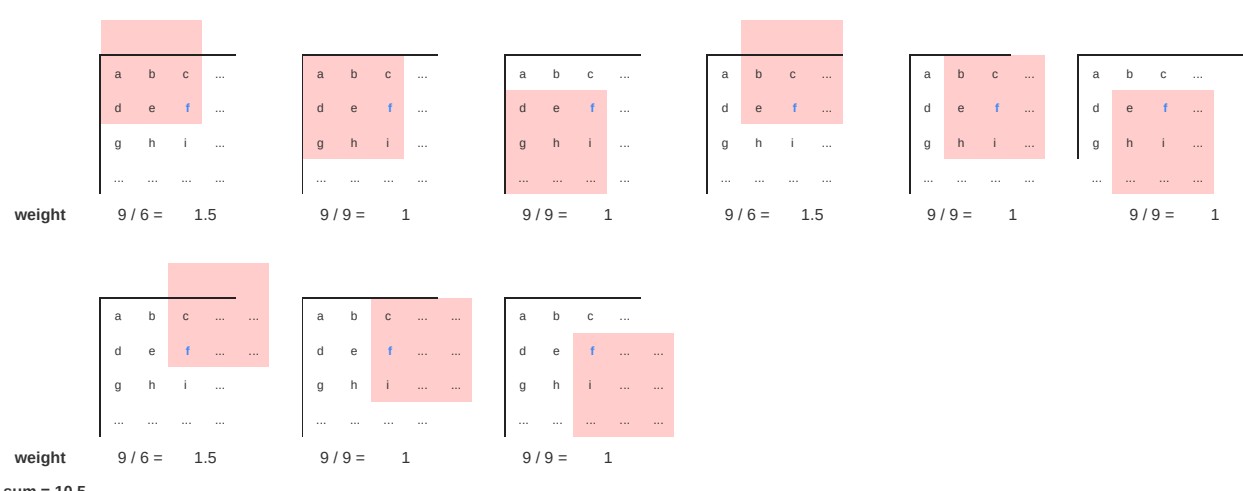

weight    9 / 6 =    1.5      9 / 9 =    1      9 / 9 =    1      9 / 6 =    1.5      9 / 9 =    1      9 / 9 =    1

weight    9 / 6 =    1.5      9 / 9 =    1      9 / 9 =    1

sum = 10.5

Convolutions involving (g): Rotated version of (c)

Convolutions involving (h): Rotated version of (f)

Convolutions involving (i): Regular uniform treatment