# OpenReview forum: "Mind the Pad -- CNNs Can Develop Blind Spots"
_ICLR.cc/2021/Conference — ICLR 2021 Spotlight_

### Official Review · AnonReviewer4 · 2020-10-28
**Relatively interesting study on elementary but often neglected shortcomings in CNN design**

**Rating:** 7
**Confidence:** 3

**Review:**

This paper addresses a commonly neglected factor in CNN design, namely the consequences of typical compromises made in padding the feature maps for convolutions and resamplings. On the downside, much of the material is known in some form in literature or practice, but on the other hand, the paper has value in systematically bringing together and analyzing these effects, and does identify and quantify effects that are apparently novel.

It is a common belief that choices like padding are minor technical details, which, even when sloppy, the network will just somehow learn to tolerate. This is to some extent true and it is remarkable how well CNN's work even when apparently crippled by questionable design. However, there is a cost and the network may need to make sub-optimal compromises to work around these issues. The paper's demonstration of lopsidedness of the learned filters is apparently novel and drives this home in an interesting way: to handle the uneven padding, the network must make a compromise where the filter kernels are slightly distorted so that they can still make a decent job of both the interior while tolerating the drastically broken boundary. However this "decent job" is not necessarily as good as it could be if the boundary defect was absent -- here the paper proposes a very simple change that demonstrably both improves performance and eliminates the kernel distortion.

I do wonder how these findings compare to e.g. the results of the explicit padding paper, which tackles these issues head on in a principle yet more complicated manner. It seems like e.g. the poor performance of standard padding in the Gaussian blur example in that paper could be explained by effects similar to the aforementioned distortion of the filters in the interior.

The paper also identifies the phenomenon of arbitrarily distorted predictions made in vicinity of padding boundaries, leading to incorrect predictions when small objects occupy this space. While unsurprising, apparently this issue has received little attention in literature, perhaps in part because e.g. imagenet data is labeled based on what is at the center of the image. In any case, higher detection accuracy is reported for many tasks when the boundary handling is improved.

In addition to these the paper analyzes the number of connections between output and input pixels when various padding strategies are used and makes some recommendations about the most balanced choices. The foveation patterns of most coarse padding methods do seem unhealthy and it's plausible that it's connected to the arbitrary nonlinear distortion in prediction scores near boundaries. While a thorough connection is not necessarily established, this is at least hinted by Fig 5b.

The paper is clearly written and easily understandable.

Overall, while the findings of the paper are arguably fairly simple and elementary compared to many works in ICLR, bringing attention to this sort of effects is valuable and I did learn something from reading the paper, so I would lean slightly towards accepting it.

_Update after rebuttal_: The authors have been very forthcoming with further analyses in the rebuttals. I believe the findings deserve to be published and that bringing attention to these issues is a good thing, and accordingly I have bumped my rating up a bit.

---

> ### Author Response · Authors · 2020-11-23
> **Explicit Boundary Handling; Foveation Patterns**
>
> We are inspired by the intuition of how CNNs “must make a compromise” to “tolerate the drastically broken boundary”.
> As proposed by our reviewer, we analyzed CNN internals under explicit boundary handling (Innamorati et al.) using the image de-bayering example for which the authors [provided their source code](http://geometry.cs.ucl.ac.uk/projects/2018/learning-edge/).
>
> Our observations are (see supplementary/rebuttal/explicit_border_handling for backing material):
>
> * The feature maps still exhibit artifacts originating from the boundaries. This is caused, in part, by potential discontinuity that takes places output when assembling the output of an interior filter and its corresponding boundary filters into one feature map.
> * The average 3x3 filter in each boundary filter set expectedly tends to resemble the corresponding boundary condition (e.g. top-left corner or bottom edge). However, the average interior filter still exhibits asymmetries which is likely influenced by the above-mentioned discontinuity.
>
> Nevertheless, the method can serve as a “learned” padding scheme if the discontinuity issue is addressed, e.g. by incorporating continuity in the learning objective of the boundary filters or by employing similar ideas as in the recently proposed [Continuous Convolution Layer](https://arxiv.org/abs/2006.11120). As such, it deserves further investigation as a competitive approach.
>
> Regarding foveation patterns, we do think those should be factored in the decision when choosing a padding method for a specific problem. **We did observe that a uniform foveation map can have a positive influence on performance**. For example, mirror padding performs consistently better under ```SYMMETRIC``` mode (uniform foveation map) than under ```REFLECT``` mode (non-uniform map) [in the experiments conducted by Liu et al](https://github.com/NVIDIA/partialconv#comparison-with-zero-padding-reflection-padding-and-replication-padding-for-5-runs) on five different ImageNet classifiers. (Note that ```REPLICATION``` is equivalent to ```SYMMETRIC``` mirror padding in all of those experiments because the padding amount is only 1 pixel which is applied around the feature maps to warrant ```SAME``` padding with 3x3 filters).\
> However, besides dictating the foveation behavior, the padding method directly impacts the formation of feature map artifacts. This can have a stronger influence on the outcome, depending on the characteristics of the task and the dataset at hand. Therefore, **we could not make a claim about which padding method is universally superior as we summarized in Section 8**.
>  Note that:
> * The foveation maps are computed under constant weights (for the purpose of *counting* input-output paths), and hence the connection between foveation effects and CNN internals is more nuanced in trained networks than what these maps suggest.
> * There are subtle nuances in the uniformity of foveation maps as we illustrated in appendix E in the paragraphs titled “Which kernel cells these ops utilize?” (cf. the illustrations for circular and symmetric mirror padding).
>
> On a similar note, applying no padding (VALID mode) diminishes the contribution of large parts of the input, as evident in its foveation map, which drastically impacts the accuracy of standard CNNs. However, such limitations might potentially be possible to overcome if the CNN is used in combination with a suited attention mechanism. Specifically, future work can investigate the use of padding-free CNNs as building blocks in [Vision Transformers](https://openreview.net/forum?id=YicbFdNTTy) to mimic foveation + eye gazes in human vision, while categorically eliminating padding-induced artifacts.
>
> We appreciate the suggestions by our reviewer which open avenues to learn "smooth" boundary handling methods and to analyze more closely how foveation patterns impact CNN internals.

---

### Official Review · AnonReviewer2 · 2020-10-29
**This submission describes how 0-padding impacts small object detection and presents how to mitigate the problem.**

**Rating:** 8
**Confidence:** 4

**Review:**

Summary of Submission:

This submission points out that 0-padding in convolutional networks induces areas where responses for object detections are significantly decreased. This leads to misdetections of small objects. The submission studies the case of traffic lights. Analysis of filter response maps at various layers of the network and analysis of the average filter weights show the effect of the 0-padding. Making the padding symmetric and using mirror or reflection padding are shown to tackle the issue.

Update after Rebuttal:
I'd like to thanks the authors for addressing the comments really thoroughly. The additional experiments using anti aliased networks are indeed very interesting. I agree with AnonReviewer4 that some of the material in this submission is probably known to people already. However, I do strongly believe that many people are unaware of the impact and importance of these effects and therefore I believe publication of this content is important.

Strengths:

- Padding is omni-present in modern CNN architectures and its effect is not often analyzed. This submission contains a thorough analysis, points out the problem with 0-padding and shows how various alternatives show significant improvement.

- Especially the analysis of the asymmetric padding that leads to asymmetric filters  in Section 5 is very interesting.

- The text is well written and easy to follow.


Weaknesses:

- The introduction describes how the issue with the 0-padding leads to drastic differences in object detection between frames in a video. There has been work [39] which also showed that aliasing in CNNs causes these effects which does not seem to be discussed in this manuscript. What would happen to the effects of the 0-padding if an anti-aliased network would be used?

- The experiments are focused on traffic lights. Given the generality of the issue presented in the manuscript I feel that additional experiments with other small objects or related problems could have been included.


Explanation of Rating:

This submission describes a problem with 0-padding that significantly impacts an important problem and shows how to mitigate the situation. Therefore I recommend accepting this submission.


Minor Comment:

Page 5: layer in RseNet -> ResNet

---

> ### Author Response · Authors · 2020-11-23
> **Antialiased CNNs; Further Examples**
>
> We thank our reviewer for encouraging us to study the relation between aliasing and padding effects.
> We compared CNN internals between baseline ResNet / VGG models and their antialiased counterparts [made available by Zhang et al](https://github.com/adobe/antialiased-cnns).
> Our observations are (see supplementary/rebuttal/antialiased_cnns for backing material):
> * While the feature maps inevitably develop spatial artifacts under zero padding, we found that **antialiasing generally smoothens these artifacts, in particular, line artifacts became less prominent**.
> * **Interestingly, the asymmetry of certain average filters in ResNet and MobileNet models is reduced in the antialiased versions**. This asymmetry is rooted in the uneven application of 0-padding during downsampling, induced by training the models on 224x224 images. Antialiasing seems increasingly able to mitigate this effect at deeper layers, where the asymmetry practically vanishes in all six models we analyzed. **This is in line with the shift-consistency maps by Zhang (ICLR 2019)** that demonstrate how antialiased models remain shift robust at deeper layers unlike the baseline models. **We think this ability is explainable by how BlurPool smoothens sharp edges** during downsampling, in turn, reducing the overrepresentation of left and top zero edges in the presence of uneven padding - we will point to the need to examine these effects in future work.
> * Intrigued by the previous point, **we quantified the shift consistency of ImageNet models we trained on 225x225 images** using the metric proposed by Zhang (ICLR 2019). **The consistency did improve for all of these models**, compared with the baseline ones trained on 224x224 images. This suggests that the asymmetry in the filters makes the model less shift-robust and can be, in part, the underlying mechanism which the model uses to implicitly encode / overfit spatial information.
>
> We will expand the discussion section to report the above initial insights and to invite further analysis of the synergies between padding and aliasing effects in CNNs.
>
> We selected traffic-light detection mainly to demonstrate how spatial artifacts look like and to qualitatively illustrate how they can mute small object detection. We included a collection of examples of these artifacts in multiple domains (see supplementary/further_CNN_examples) and demonstrated a similar interference in the face detection example. We added one more example to that collection to showcase the impact of padding artifacts in image inpainting and to demonstrate significant qualitative improvement after training under mirror padding.
> To analyze the effects of unevenly-applied padding, we opted for ImageNet classification with ResNet / MobileNet models for the following reasons:
> * Standard **ResNet / MobileNet models do perform zero padding at downsampling layers**, the culprit of potential asymmetry, and  widely used as "trunk" models in practice, making the analysis potentially relevant to a broad audience.
> * The standard **training size** is *fixed* to 224x224 which **gives rise to unevenly-applied 0-padding at *every* downsampling layer** of these models. Mitigating it only requires increasing the size to 225x225, hence avoiding excessive padding that would dilute the effect we are after. In contrast, RCNN-based detectors are trained on COCO using *variable* input size to leverage the full resolution of the training data. This dilutes the effect we are after because 1) the chance to unevenly apply 0-padding is statistically reduced by 50%, and 2) rectifying arbitrary input sizes to warrant even application of padding at all layers can require excessive input padding.
> * Both **the nature of a classification task and the image statistics of ImagNet arguably encourage the emergence of symmetric mean filters**. This might not be the case in Chinese character classification, where vertical strokes dominate the training data, or in drivable-area detection, where roads dominate certain parts of traffic scenes.
> * ImagNet classification serves as **a versatile testbed** that makes it easy to demonstrate the effects we measure and to compare
>  them with other effects (e.g. aliasing). The nuances of more involved computer-vision tasks (e.g. inpainting masks) make it a bit harder to isolate these effects.
>
> We nevertheless hinted about the need "to closely examine the implications of spatial bias and foveation in various applications", perhaps in dedicated studies within ML sub-communities, and tried to focus on generic learning-representation aspects and their practical implications.

---

### Official Review · AnonReviewer1 · 2020-10-29
**Quite empirical analysis (rating raised after rebuttal)**

**Rating:** 6
**Confidence:** 3

**Review:**

Summary: This paper does empirical analysis on an object detector, esp. how it fails due to slight shift of objects in videos. By using different padding schemes for conv filters, the authors find the padding scheme is the root for such failures.

Update after rebuttal:
The author responses and manuscript revisions have addressed most of my concerns. Now I lean towards acceptance.

Reasons for score:
1. This paper is not written professionally. The introduction is very short. The following sections are more like experimental notes than a technical paper.
2. The analysis is mostly based on a single use case, i.e. an SSD object detector for traffic lights. This study is highly insufficient. More use cases and application scenarios should be investigated.
3. Experimental evaluation is also highly insufficient. Only changing the input image size by 1 pixel doesn't reveal much. A lot of ablation studies and changing of padding schemes should be evaluated, preferably on a few datasets of different tasks (classification, detection, segmentation, etc.).

Other comments:
I agree that padding could lead to biases when doing downsampling using stride > 1, as illustrated in Fig. 6. However, this seems to be easily fixed by doing random image flipping as one of the data augmentations? It would even this bias. So this issue may not be practically severe, although it is worth study per se.

Minor issues:
The PDF file of this paper seems to have issues. The printed characters are blurry, and acrobat reports errors when open it. But some other PDF readers can open it.

---

> ### Author Response · Authors · 2020-11-24
> **Ablation Studies; Application Scenarios; Random Image Flipping**
>
> We thank our reviewer for proposing further ablation studies.
> Indeed, we repeated the ImageNet experiments on two of the ResNet models, this time by expanding the input dimensions by 2 pixels (using mean padding), from 224x224 to 226x226. The improvement in top-1 (and top-5) accuracy matched the one we observed when expanding the input by 1 pixel only (from 224x224 to 225x225):
>
> |Input Size|ResNet-18|ResNet-34|
> |---|---|---|
> 224 × 224|69.93% (89.22%)| 73.30% (91.42%)|
> |225 × 225|70.27% (89.52%)|73.72% (91.58%)|
> |226 × 226|70.07% (89.52%)|73.59% (91.57%)|
> |256 × 256|70.24% (89.52%)| N/A|
>
> **This supports our claim that the boost is probably due to eliminating uneven application of padding and not merely due to the increase in input size**. To further verify this observation, we expanded the input dimensions from 224x224 to 256x256 by applying 16-pixel mean padding around the input images. Training ResNet-18 on these images did not give any advantage over 225x225 inputs.
>
> > changing of padding schemes should be evaluated
>
> We did experiment with changing the padding method in ResNet-18 from zero to circular padding and training both on 224x224 images and on 225x225 images.
> In both cases, the accuracy improved visibly under circular padding:
>
> |Padding Method|Training on 224x224 images| Training on 225x225 images|
> |---|---|---|
> |Zero Padding|69.93% (89.22%)|70.27% (89.52%)|
> |Circular Padding|70.28% (89.44%)|70.47% (89.80%)|
>
> This is expected since circular padding significantly reduces feature-map artifacts that emerge under zero padding due to injecting zeros in the padding areas, as we explained in Section-7 and elaborated in Appendix D.
> Furthermore, training under circular padding also mitigates the asymmetry in mean filters, even when training on 224x224 images (see Appendix C). This is because even when the padding is applied only at the left and top borders, padding by reusing the feature-map values prevents the disproportional presence of zeros at these borders, compared with the right and bottom borders.
>
> We included a similar analysis under mirror padding in the SSD example (Section 7).
> **Our purpose is not to establish the superiority of a specific padding method**, we did argue that the choice is problem dependent (Section 8). For example circular padding does not interfere much with ImageNet classification, since the objects tend to be mostly in the center. On the other hand, it might be less suited in small-object detection as it makes neighbors out of distant pixels (hence our choice of mirror padding in the SSD experiment). **Instead, we wanted to raise awareness about potentially harmful effects of padding in CNNs (with a broader ML audience in mind), to exemplify how these could be mitigated, and to aid domain experts with a "checklist" of aspects to pay attention to when applying padding**.
>
> > preferably on a few datasets of different tasks (classification, detection, segmentation, etc.).
>
> In our response to AnonReviewer2, we elaborated on the rationale behind choosing ImageNet classification and traffic-light detection to showcase our analysis and findings.
>
> > The analysis is mostly based on a single use case [...] More use cases and application scenarios should be investigated.
>
> We did include a collection of examples demonstrating CNN spatial bias in multiple domains, including semantic segmentation, image colorization and inpainting, reinforcement learning, and spectrogram-based audio classification (see supplementary/further_CNN_examples).
>
> > The introduction is very short. The following sections are more like experimental notes than a technical paper.
>
> We renamed the first section to “Motivation” to better reflect its content.
> Since our contribution revolves around edge cases and subtle effects in CNNs, we opted to first illustrate these cases and effects (Sect. 2-3), then highlight why padding is important despite being the root cause (Sect. 4), and finally demonstrate how applying padding carefully can mitigate these effects (Sect. 5-7) based on theoretical analysis, together with empirical results.
>
> > this seems to be easily fixed by doing random image flipping as one of the data augmentations?
>
> Random horizontal flipping is indeed part of the ImageNet training recipe in PyTorch, which we used in our experiments. However, it does not fix the issue since the uneven application of padding (and the bias it incurs) takes place, more importantly, "inside the CNN" around the feature maps that are fed into downsampling layers. Randomly flipping feature maps is very likely to impede the learning - a dedicated study is needed to evaluate this issue. (Note that both vertical and horizontal flipping would be needed to even out the bias).
>
> Finally, we thank our reviewer for reporting issues with viewing the manuscript. We will update the draft as explained above and make sure it can be viewed properly in different PDF viewers.

---

### Official Review · AnonReviewer3 · 2020-11-04

**Rating:** 8
**Confidence:** 4

**Review:**

The paper studies the effect of padding on artefacts in CNN feature maps and performance on image classification and object detection. It convincingly makes the case that these artefacts have a significant detrimental effect on task performance, e.g. leading to blind spots / missed detections of small objects near the image border. It also studies the effect of uneven padding in downsampling layers, where the padding may only affect some sides of the image and not others, depending on the image size. A condition is presented for when this does / does not occur. The effect of different padding methods is also studied from the perspective of foveation by computing the number of paths from an input pixel to the output. A number of practical recommendations are given.

The paper is well written. It contains lots of details that are relevant to CNN architecture design, especially when the appendix is taken into account. Proposed fixes are simple and produce a very significant improvement in performance on imagenet classification and object detection, so are likely to be adopted by practitioners.

The paper states:
"It is evident that the 1-pixel border variations in the second map are caused by the padding mecha-nism in use. This mechanism pads the output of the previous layer with a 1-pixel 0-valued border inorder to maintain the size of the feature map after applying a 3x3 convolutional kernel. The maps inthe first layer are not impacted because the input we feed is zero valued. Subsequent layers, however,are increasingly impacted by the padding, as preceding bias terms do not warrant 0-valued input."
It would be interesting to know if batchnorm or some other kind of normalization might mitigate this issue, because if the feature map is constant but non-zero, normalization will make it all zero. Of course this will not hold for non-zero (natural) inputs, but it would still be interesting to see a discussion on the effect of (batch) normalization on padding artefacts.


Typos
Section 4: "To serves"
Section 5: RseNet

------
UPDATE
I have now read the other reviews, author response and updated paper, and have decided to maintain my rating.

---

> ### Author Response · Authors · 2020-11-24
> **batchnorm**
>
> We thank our reviewer for bringing up the impact of normalization.
> Indeed, while spatial bias is induced by the padding mechanism, its *magnitude* is impacted by normalization:
> * As our reviewer noted, `batchnorm` does not harmonize the "content" (i.e. the individual pixels) of a feature map, instead, it scales and shifts the map as a whole by learned factors.
> As such, `batchnorm` does not prevent the emergence nor the propagation of spatial bias.
> * However, `batchnorm` tends to limit and harmonize the range of variation across different feature maps (see supplementary/rebuttal/batchnorm for comparison between baseline VGG models and their `batchnorm` versions).
> * This, in turn, impacts how possible artifacts in these maps accumulate when they are used in the next layer (see Figure 4 in the manuscript for illustration).  Ideally, we want these artifacts to cancel out, on average. This is not warranted as it is not part of the training objective.
> * Moreover, `batchnorm` is  usually followed by crucial ReLU units, as in the case of MobileNet and ResNet models. All artifacts that manifest in the feature maps after applying ReLU are of a positive sign, and are hence less likely to cancel out in the next conv layer.
>
> A closely-related issue to padding effects are aliasing effects in CNNs, brought up by AnonReviewer2.
> By examining [the accuracy improvements antialiasing achieves in various models](https://github.com/adobe/antialiased-cnns#3-imagenet-evaluation-results-and-training-code), we found that VGG models that use `batchnorm` (vgg11_bn, vgg13_bn, vgg16_bn, and vgg19_bn) benefited significantly more from antialiasing than those that do not (vgg11, vgg13, vgg16, and vgg19), specifically (+2.25 **vs.** +1.49, +2.06 **vs.** +1.59, +1.77 **vs.** +1.37, and +1.44 **vs.** +1.16). These models do not benefit much from our proposed input size adjustments because they avoid padding during downsmapling (see appendix C - note on VGG models).  However, it would be very interesting to design experiments specifically to study possible interaction between padding effects and normalization.
>
> We hope the above sheds some light on such interaction. We will add a paragraph to the discussion section to bring up this issue and to invite further work to analyze this interaction under different normalization and padding mechanisms. Finally, besides normalization, further architectural aspects can impact the magnitude and / or the propagation of spatial bias, such as filter size and dilation factors.

---

### Decision · Program_Chairs · 2021-01-07
**Final Decision**

**Decision:**

Accept (Spotlight)

**Comment:**

This submission explores how certain common padding choices can induce spatial biases in convolutional networks. It looks into alternative padding schemes which mitigate these issues and demonstrates significant performance improvements in widely used convnets. Reviewers generally agreed that this is an important point that should be more widely understood in the community, and that the proposed changes are relatively simple to adopt, so this work is likely to be impactful. Most reviewers thought the paper was well-written, describing the problem well, and the analysis well-executed. Most reviewers acknowledged that most of the weaknesses described in their initial reviews were well-addressed by the authors' responses and manuscript updates. Given the strength of the analysis and the impact for many practitioners, I recommend the submission be accepted with a spotlight presentation.